# Learning to Instruct for Visual Instruction Tuning

**Zhihan Zhou**[*1]  **Feng Hong**[*1]  **Jiaan Luo**[1]  **Yushi Ye**[1]
**Jiangchao Yao**[⊠1]  **Dongsheng Li**[2]  **Bo Han**[3]  **Ya Zhang**[4,5]  **Yanfeng Wang**[4]
[1]Cooperative Medianet Innovation Center, Shanghai Jiao Tong University
[2]Microsoft Research Asia  [3]Hong Kong Baptist University
[4]School of Artificial Intelligence, Shanghai Jiao Tong University
[5]Institute of Artificial Intelligence for Medicine, Shanghai Jiao Tong University School of Medicine
{zhihanzhou, feng.hong, Sunarker}@sjtu.edu.cn

## Abstract

We propose L2T, an advancement of visual instruction tuning (VIT). While VIT equips Multimodal LLMs (MLLMs) with promising multimodal capabilities, the current design choices for VIT often result in overfitting and shortcut learning, potentially degrading performance. This gap arises from an overemphasis on instruction-following abilities, while neglecting the proactive understanding of visual information. Inspired by this, L2T adopts a simple yet effective approach by incorporating the loss function into both the instruction and response sequences. It seamlessly expands the training data, and regularizes the MLLMs from overly relying on language priors. Based on this merit, L2T achieves a significant relative improvement of up to 9% on comprehensive multimodal benchmarks, requiring no additional training data and incurring negligible computational overhead. Surprisingly, L2T attains exceptional fundamental visual capabilities, yielding up to an 18% improvement in captioning performance, while simultaneously alleviating hallucination in MLLMs. Github code: https://github.com/Feng-Hong/L2T.

## 1 Introduction

Large Language Models (LLMs) have achieved significant progress and success. Built on this, Multimodal LLMs (MLLMs) have garnered substantial attention in the research community. There has been a surge in advancements based on the Visual Instruction Tuning (VIT) [Liu et al., 2023a] paradigm. By aligning language inputs and visual representations through simple connectors, and subsequently conducting end-to-end fine-tuning on carefully designed multimodal instruction data, MLLMs [Liu et al., 2023a, 2024a,b, Zhu et al., 2024, Tong et al., 2024] have achieved notable improvements across various multimodal tasks like visual question answering and image captioning.

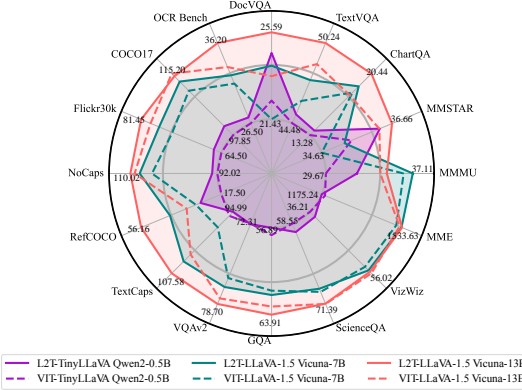

Figure 1: Performance comparison on 16 tasks between L2T and VIT using different models, including TinyLLaVA Qwen2-0.5B [Zhou et al., 2024], LLaVA-1.5 Vicuna-7B [Liu et al., 2024a], and LLaVA-1.5 Vicuna-13B [Liu et al., 2024a]. The pretraining phase uses the LLaVA-pretrain-558k dataset, while the fine-tuning phase employs the LLaVA-mix-665k dataset.

---

* Equal contribution. Work done during Feng Hong's internship at Microsoft Research Asia.

39th Conference on Neural Information Processing Systems (NeurIPS 2025).

Typically, visual instruction tuning consists of two stages: pre-training and fine-tuning [Liu et al., 2023a, Zhu et al., 2024]. In the pre-training stage, a connector is trained to align visual and language data. The fine-tuning phase is similar to instruction tuning in language models, where the model is trained to generate responses based on multimodal instructions. However, recent research has shown that instruction tuning can lead to a series of issues, such as the knowledge degradation [Ghosh et al., 2024] and hallucinations [Rawte et al., 2023, Tonmoy et al., 2024]. Similar problems have also been observed in models that employ visual instruction tuning [Huang et al., 2023, Leng et al., 2024]. The potential causes may lie in overfitting and shortcut learning during instruction tuning [Sun et al., 2024a]. Figure 2 illustrates an example of a shortcut, where the model might ignore visual content and generate responses based solely on language priors. Current advancements [Liu et al., 2024a,b, Bai et al., 2023b, Tong et al., 2024] in MLLMs implicitly mitigate such issues by leveraging larger, higher-quality, and more diverse training datasets, larger models, and improved pretrained initializations for visual and language backbones.

In this paper, we propose an orthogonal solution to improving visual instruction tuning: learning to instruct images as a regularizer, which we refer to as **L**earning **to** Instruc**T** (L2T). Specifically, in addition to learning to generate responses to given images and instructions as usual, L2T also learn to generate instructions for images that exclude templates, which refer to special tokens and high-frequency, low-information template tokens in the instructions. L2T enhances VIT by: (1) expanding the training content to mitigate overfitting without explicitly enlarging the training set (Section 3.4); and (2) learning to instruct images, which forces the model to focus more on the visual content and prevents learning some shortcuts (Section 2).

We conduct extensive experiments across a range of 16 tasks, comparing L2T and VIT on different models. As shown in Figure 1, L2T achieves a significant relative improvement of up to 6% over VIT in overall multimodal task performance. Moreover, L2T shows significant potential in alleviating hallucination issues in MLLMs, as demonstrated across four diverse hallucination benchmarks (Section 3.3). Through comprehensive and detailed experiments, L2T demonstrates the following advantages: (1) It improves the ability across different multimodal tasks compared to the conventional visual instruction tuning, especially those focusing more on visual content, such as OCR, captioning and hallucination mitigation. (2) Our method is orthogonal to existing research advancements and can further enhance MLLMs' performance through simple integration. (3) It shows advantages in improving performance while making minimal compromises in training and inference costs.

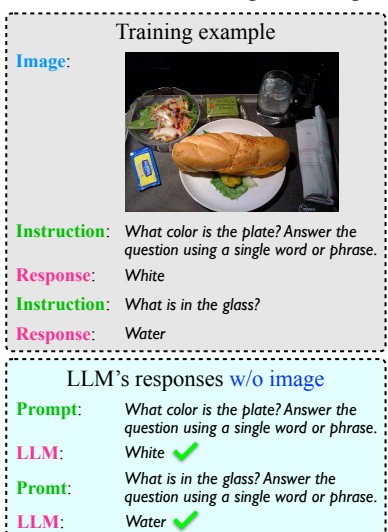

Figure 2: An example where a pure language model provides correct answers based only on language priors, without relying on visual content. This shows that learning to generate responses alone cannot prevent the model from taking shortcuts by ignoring visual content and relying solely on textual instructions.

## 2  Method

**Problem Formulation.** Taking LLaVA as an example, a typical model architecture comprises a pre-trained large language model $f(\cdot)$ with parameters $\theta_f$ with the corresponding tokenizer and embedding layer $t(\cdot)$ (*e.g.*, Vicuna [Peng et al., 2023]), a pre-trained visual feature extractor $g(\cdot)$ with parameters $\theta_g$ (*e.g.*, CLIP-ViT-L/14 [Radford et al., 2021]), and a cross-modal connector $h(\cdot)$ with parameters $\theta_h$, such as a linear layer or MLP. Let $\theta = \{\theta_f, \theta_g, \theta_h\}$ denote the set of all parameters. For an image $\mathbf{X}_V$ and a related text instruction $\mathbf{X}_I$, we obtain the corresponding textual response through the following forward process. As shown in Figure 3, the image $\mathbf{X}_V$ is forwarded through the visual feature extractor $g$ and cross-modal connector $h$, mapping it to visual tokens $\mathbf{H}_V = h(g(\mathbf{X}_V))$ in the language embedding space. $\mathbf{H}_V$ is then combined with the language tokens $\mathbf{H}_I = t(\mathbf{X}_I)$ to form a sequence, which is forwarded into LLM $f$ for autoregressive generation of the response sequence $\mathbf{X}_A$. The goal of visual instruction tuning is to train the model to exhibit strong multi-modal instruction-following capabilities.

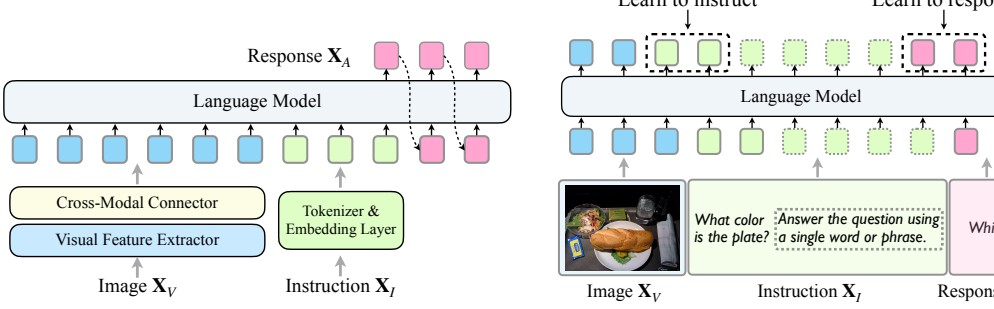

Figure 3: The model architecture using LLaVA as an example, and the data flow for generating responses from images and instructions.

Figure 4: Illustration of L2T. In addition to learning to generate responses like VIT, L2T also learns to generate instructions that exclude templates.

**Visual Instruction Tuning (VIT).** For a given training sample triplet $(\mathbf{X}_V, \mathbf{X}_I, \mathbf{X}_A)$[1], the sequence lengths of $\mathbf{X}_I$ and $\mathbf{X}_A$ are $L_I$ and $L_A$, respectively. The training objective of VIT is to learn to generate the response $\mathbf{X}_A$. Specifically, it involves learning to predict each token $\mathbf{X}_{A,i}$ in $\mathbf{X}_A$, where $i \in \{1, 2, \ldots, L_A\}$, based on the image $\mathbf{X}_V$, the instruction $\mathbf{X}_I$, and the preceding response sequence $\mathbf{X}_{A,<i}$. The loss function $\mathcal{L}$ for the training sample is formulated as the negative log-likelihood of the response given the image and the instruction:

$$\mathcal{L} = -\log p_\theta(\mathbf{X}_A|\mathbf{X}_V, \mathbf{X}_I) = -\sum_{i=1}^{L_A} \log p_\theta(\mathbf{X}_{A,i}|\mathbf{X}_V, \mathbf{X}_I, \mathbf{X}_{A,<i}). \tag{1}$$

Training is typically divided into two stages: pretraining and fine-tuning. In the pretraining stage, only the cross-modal connector is trained to align visual features with the language embedding space. In the fine-tuning stage, the entire model is trained end-to-end, with options to freeze the visual feature extractor and apply LoRA [Hu et al., 2022] fine-tuning.

**Learning to Instruct (L2T).** Building upon VIT, we propose extending its paradigm through learning to instruct images. Specifically, in addition to learning how to generate a response given an image and an instruction, we also learn how to generate a meaningful instruction for a given image. For a training sample triplet $(\mathbf{X}_V, \mathbf{X}_I, \mathbf{X}_A)$, we define the loss function $\mathcal{L}$ as the negative log-likelihood of both the instruction and the response conditioned on the image:

$$\mathcal{L} = -\log p_\theta(\mathbf{X}_I, \mathbf{X}_A|\mathbf{X}_V) = \underbrace{-\sum_{i=1}^{L_I} \log p_\theta(\mathbf{X}_{I,i}|\mathbf{X}_V, \mathbf{X}_{I,<i})}_{\text{Learn to Instruct}} \underbrace{-\sum_{i=1}^{L_A} \log p_\theta(\mathbf{X}_{A,i}|\mathbf{X}_V, \mathbf{X}_I, \mathbf{X}_{A,<i})}_{\text{Learn to Respond}}.$$

$$\tag{2}$$

By learning to generate appropriate instructions for images, L2T achieves two key benefits. 1)It naturally expands the data the model learns to fit, helping to alleviate potential overfitting to some extent. 2) Learning to instruct images ensures that the model focuses on the image content, effectively preventing it from ignoring the visual input and relying solely on languages to generate responses.

**Template Removal.** To ensure the model learns meaningful content related to the image, we exclude the learning of certain irrelevant parts of instructions. Such irrelevant context primarily arises from two sources: (1) system templates and (2) task templates. Specifically, system templates refer to the tokens used to guide the MLLMs in adopting the role of a helpful and polite AI assistant, or as the conversational clues distinguishing whether the content is generated by the "USER" or the "ASSISTANT". System templates can be easily removed when it is added. Task templates refer to tokens that indicate the task type and output format. We identify task templates by calculating the frequency of all sentences in the entire training dataset and selecting the most frequent ones. The detailed removed task templates can be referred to Appendix A. Note that the training data in the

---

[1] For simplicity, we use single-turn conversations as an example, which can be naturally extended to multi-turn conversations.

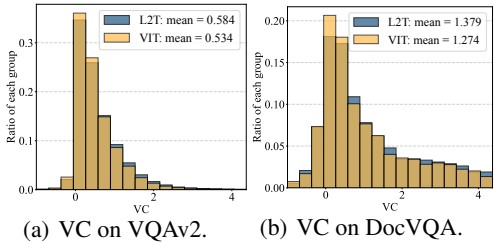
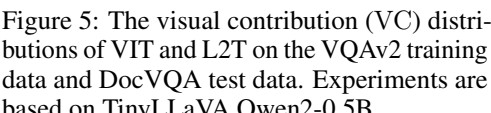

(a) VC on VQAv2.    (b) VC on DocVQA.

Figure 5: The visual contribution (VC) distributions of VIT and L2T on the VQAv2 training data and DocVQA test data. Experiments are based on TinyLLaVA Qwen2-0.5B.

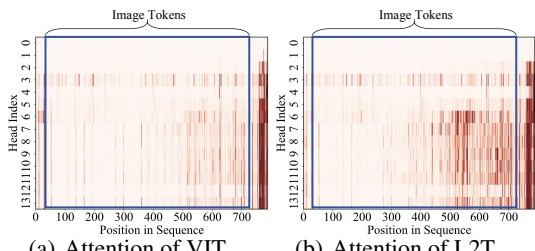

(a) Attention of VIT.    (b) Attention of L2T.

Figure 6: Visualization of attention weights. Darker colors indicate higher attention weights. Experiments are based on TinyLLaVA Qwen2-0.5B.

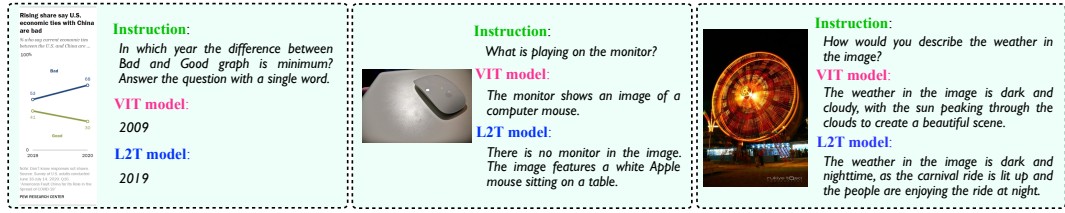

Figure 7: Cases, each including an image, an instruction, and the responses generated by the VIT and L2T models, respectively. The models are based on LLaVA-1.5 Vicuna-7B. (Left) The VIT model shows better OCR capabilities. (Middle) The VIT model demonstrates robustness and effectively avoids being influenced by misleading information in the language instruction. (Right) The VIT model provides a more comprehensive and accurate image description.

pretraining stage is constructed using image-caption pairs, with a set of template prompts serving as instructions, which are unrelated to the image content. Therefore, we apply our method L2T only during the end-to-end fine-tuning stage. In Figure 4, we present an illustration of L2T.

**Analysis.** To assess the rationality of L2T in alleviating shortcut learning in MLLMs, we conduct experiments to gain deeper insight into which aspects of visual or instructional content are most influential in response prediction. Specifically, we first introduce visual contribution (VC), which quantifies the difference in the log-likelihood of the response when conditioned on both the image and instruction, versus when conditioned solely on the instruction:

$$\text{VC} = \log p_\theta(\mathbf{X}_A|\mathbf{X}_V, \mathbf{X}_I) - \log p_\theta(\mathbf{X}_A|\mathbf{X}_V = \emptyset, \mathbf{X}_I)$$

where $\mathbf{X}_V = \emptyset$ in the implementation denotes that the original visual input is replaced with random noise. In this spirit, VC captures the relative importance of visual signals in response prediction, isolating the contribution of visual information from any confounding effects due to the instructional signals. Figure 5 presents the VC of L2T and VIT for both the training and testing datasets. Our findings reveal that L2T achieves a substantial relative improvement, with a 9% increase in VC, highlighting its effectiveness in regularizing MLLMs to better utilize visual inputs.

To further substantiate these findings, we visualize the attention weights in L2T and VIT. As shown in Figure 6, L2T leads to stronger activation in the visual components of MLLM attention heads, indicating more effective utilization of visual inputs. This is consistent with the VC results. More experimental details and additional visualization results can be referred to Appendix F.

Moreover, Figure 7 presents several cases, each consisting of an image, an instruction, and the responses generated by the VIT and L2T models, respectively. It can be observed that by learning to instruct images, the L2T model demonstrates more precise and comprehensive image understanding, as well as stronger robustness against misleading information in the language instructions.

# 3 Experiment

## 3.1 Setup

**Model Architectures.** (1) TinyLLaVA [Zhou et al., 2024]: We follow TinyLLaVA to utilize small-scale LLMs, including Qwen-2-0.5B [Yang et al., 2024] and Phi-2-3B [Gunasekar et al., 2023]. The SigLIP-400M [Zhai et al., 2023] is adopted as the vision encoder due to its effective performance in combination with small-scale LLMs. The multimodal connector follows LLaVA 1.5 Liu et al. [2024a] with a two-layer MLP and GELU activation. (2) LLaVA 1.5, LLaVA-NeXT [Liu et al., 2024b]: For LLaVA 1.5, we use Vicuna-v1.5-7B [Peng et al., 2023] and Vicuna-v1.5-13B [Peng et al., 2023] as the base LLMs, while for LLaVA-NeXT, Vicuna-v1.5-7B serves as the base LLM. The base vision encoder is CLIP-ViT-L-14 [Radford et al., 2021], and the connector is a two-layer MLP.

**Training Datasets.** (1) Pretraining Stage: We adopt the LLaVA 1.5 framework, utilizing the LLaVA-pretrain-558k data [Liu et al., 2024a] for all pretraining phases. (2) Finetuning Stage: For instruction tuning, we use the LLaVA-mix-665k data [Liu et al., 2024a] on TinyLLaVA and LLaVA 1.5. For LLaVA-NeXT, we use the LLaVA-NeXT-Data [Liu et al., 2024b], an expansion of LLaVA-mix-665k with diverse instruction data. More details can be referred to Appendix B.2.

**Implementation Details.** We train all models on NVIDIA A100 GPUs, strictly following the training recipes of TinyLLaVA, LLaVA 1.5 and LLaVA-NeXT. See Appendix B.1 for more details.

**Evaluation Benchmarks.** We conduct a thorough evaluation of four distinct capabilities using 16 multimodal instruction datasets. These capabilities are categorized as follows: (1) General Visual Question Answering, assessed through VQAv2 [Goyal et al., 2017], GQA [Hudson and Manning, 2019], ScienceQA [Lu et al., 2022], and VizWiz [Gurari et al., 2018]; (2) Comprehensive Multimodal Benchmarks, evaluated using MME [Fu et al., 2023], MMMU [Yue et al., 2024], and MMStar [Chen et al., 2024b]; (3) Chart, Document, and OCR Understanding, evaluated using ChartQA [Masry et al., 2022], TextVQA [Singh et al., 2019], DocVQA [Mathew et al., 2021], and OCR Bench [Liu et al., 2023b]; and (4) Image Captioning, assessed through COCO2017 [Lin et al., 2014], Flickr30k [Young et al., 2014], NoCaps [Agrawal et al., 2019], RefCOCO [Kazemzadeh et al., 2014], and TextCaps [Sidorov et al., 2020]. The performance metrics used include CIDEr [Vedantam et al., 2015] for all captioning tasks, the perception score for MME, and accuracy for the remaining tasks. The experiments are conducted using LMMs-Eval[2] [Zhang et al., 2024]. In Section 3.4, some tasks are evaluated on the lite version, denoted as "-L".

## 3.2 Main Results

In Table 1, we present a comprehensive evaluation of L2T across multiple benchmarks.

**General Visual Question Answering.** We first evaluate several widely used VQA benchmarks, including the most prominent VQAv2 benchmark and the visual reasoning-oriented GQA. We also assess VizWiz, which requires predicting unanswerable questions, and ScienceQA, which covers a diverse range of scientific topics. Our results demonstrate that our proposed L2T achieves competitive VQA performance when compared to VIT.

**Comprehensive Multimodal Benchmarks.** We evaluate the multi-discipline multimodal understanding and reasoning capabilities of our L2T through MME, MMMU and MMStar benchmarks. These benchmarks span a wide range of disciplines, including Art & Design, Business, Science, Health & Medicine, Humanities & Social Sciences, Technology & Engineering, and encompass diverse tasks such as object presence, counting, spatial reasoning, commonsense inference, numerical computation, and code reasoning. Our results demonstrate that L2T consistently outperforms VIT, achieving an average relative improvement of up to 3.7%. This underscores the effectiveness of L2T in enhancing performance across diverse multimodal applications.

**Chart, Document, and OCR Understanding.** We include a diverse set of OCR-related datasets to evaluate the performance of models in locating, extracting and interpreting text from various data visualizations, including charts, diagrams, documents, and natural images, in order to accurately address the associated questions. The results indicate that our L2T significantly outperforms VIT across all OCR-related benchmarks, yielding an average relative improvement of 6.3%. This high-

---

[2]https://github.com/EvolvingLMMs-Lab/lmms-eval/

Table 1: Performance comparison of VIT and L2T across 16 representative multimodal benchmarks. The benchmarks span four categories: General VQA, multi-discipline benchmarks, Chart/Document/OCR understanding, and image captioning. We report the average relative improvement for each category, along with the overall relative improvement across all benchmarks. [*]LLaVA-NeXT's training script is not fully open-sourced; our implementation may differ slightly.

| Benchmark | Visual Instruction Tuning (VIT) | | | | | Learning to Instruct (L2T) | | | | |
| --- | --- | --- | --- | --- | --- | --- | --- | --- | --- | --- |
| | TinyLLaVA Qwen-2-0.5B | TinyLLaVA Phi-2-3B | LLaVA-1.5 Vicuna-7B | LLaVA-1.5 Vicuna-13B | LLaVA-NeXT[*] Vicuna-7B | TinyLLaVA Qwen-2-0.5B | TinyLLaVA Phi-2-3B | LLaVA-1.5 Vicuna-7B | LLaVA-1.5 Vicuna-13B | LLaVA-NeXT[*] Vicuna-7B |
| VQAv2 
 General QA | 72.31 | 77.32 | 76.64 | 78.26 | 80.07 | 72.35 | 77.48 | 77.35 | 78.70 | 80.62 |
| GQA 
 General QA | 57.48 | 61.46 | 61.97 | 63.25 | 64.24 | 56.89 | 61.12 | 62.32 | 63.91 | 62.54 |
| ScienceQA 
 Science QA | 58.55 | 71.78 | 69.46 | 71.39 | 70.10 | 59.89 | 71.00 | 68.96 | 71.34 | 70.60 |
| VizWiz 
 General QA | 36.21 | 40.46 | 54.39 | 56.53 | 55.60 | 37.84 | 41.85 | 55.56 | 56.02 | 58.66 |
| **Average** | - | - | - | - | - | **+1.5%** | **+0.5%** | **+0.7%** | **+0.2%** | **+1.1%** |
| MME 
 Multi-discipline | 1189.99 | 1475.51 | 1508.26 | 1522.60 | 1477.28 | 1175.24 | 1431.12 | 1531.37 | 1533.63 | 1556.44 |
| MMMU 
 Multi-discipline | 29.67 | 36.67 | 36.33 | 34.33 | 36.11 | 32.33 | 37.89 | 37.11 | 34.89 | 35.56 |
| MMStar 
 Multi-discipline | 34.93 | 36.07 | 33.65 | 36.12 | 37.26 | 36.14 | 36.44 | 34.63 | 36.66 | 38.98 |
| **Average** | - | - | - | - | - | **+3.7%** | **+0.4%** | **+2.2%** | **+1.3%** | **+2.8%** |
| ChartQA 
 Chart Understanding | 13.28 | 16.48 | 18.24 | 17.92 | 54.88 | 13.80 | 17.20 | 18.96 | 20.44 | 66.80 |
| TextVQA 
 OCR | 44.48 | 52.10 | 46.09 | 48.73 | 64.82 | 45.12 | 52.35 | 47.58 | 50.24 | 65.06 |
| DocVQA 
 Doc. Understanding | 22.32 | 28.46 | 21.43 | 23.49 | 68.49 | 24.60 | 30.74 | 23.99 | 25.59 | 72.52 |
| OCR Bench 
 OCR | 26.50 | 34.50 | 31.30 | 33.30 | 52.00 | 27.20 | 35.40 | 32.30 | 36.20 | 55.80 |
| **Average** | - | - | - | - | - | **+4.6%** | **+3.9%** | **+5.6%** | **+8.7%** | **+8.8%** |
| COCO2017 
 Image Captioning | 97.85 | 100.82 | 110.38 | 115.20 | 99.96 | 100.37 | 102.61 | 112.96 | 114.46 | 139.21 |
| Flickr30k 
 Image Captioning | 64.50 | 76.00 | 74.85 | 79.45 | 68.46 | 66.09 | 78.34 | 77.64 | 81.45 | 75.26 |
| NoCaps 
 Image Captioning | 92.02 | 101.13 | 105.54 | 109.15 | 88.35 | 93.08 | 103.60 | 108.09 | 110.02 | 114.10 |
| RefCOCO 
 Image Captioning | 17.50 | 30.59 | 29.76 | 34.22 | 33.82 | 27.55 | 52.98 | 42.24 | 56.16 | 35.36 |
| TextCaps 
 Image Captioning | 95.80 | 105.79 | 98.15 | 103.71 | 70.40 | 94.99 | 105.36 | 105.14 | 107.58 | 74.03 |
| **Average** | - | - | - | - | - | **+12.6%** | **+16.0%** | **+11.5%** | **+14.1%** | **+17.6%** |
| **Overall** | - | - | - | - | - | **+6.1%** | **+6.2%** | **+5.6%** | **+6.9%** | **+8.5%** |

lights the effectiveness of the proposed L2T in extracting fine-grained visual information. This can be attributed to the fact that our L2T encourages the proactive generation of instructions. In OCR-related datasets like TextVQA, the instructions require predicting "Reference OCR token" that captures the informative textual content within the image. This necessitates a more comprehensive understanding of the low-level information embedded in the images, thus enhancing OCR capabilities.

**Image Captioning.** We assess image-captioning, a representative image-text task widely used as a pretraining task for MLLMs. Our analysis includes classical captioning datasets such as COCO2017 and Flickr30k, along with more specialized datasets such as TextCaps, which involves fine-grained textual content description, and NoCaps, which represents out-of-domain scenarios. The results demonstrate that L2T achieves the most substantial performance improvement over VIT on captioning datasets, among the various capabilities evaluated, with an average relative improvement of up to 17.6%. The reasons are two-fold: (1) L2T motivates a more comprehensive understanding of the images, which guarantees the generation of accurate descriptions, and (2) L2T alleviates the negative effects of overfitting on instruction-following abilities, preserving more of the captioning skills acquired during the pretraining stage.

**Finding.** L2T consistently outperforms VIT across all benchmarks, achieving an average relative improvement of up to 8.5%. The most significant gains are observed in OCR-related and captioning datasets, with improvements of up to 8.8% and 17.6%. This suggests that L2T excels at regularizing MLLMs to pay more attention to visual input and strengthening their fundamental visual capabilities.

### 3.3 Hallucination Evaluation

In this section, we evaluate the effectiveness of L2T in mitigating hallucinations, as shown in Table 2.

Table 2: Hallucination evaluation results. We present the average accuracy computed across adversarial, popular and random splits of POPE, along with per-instance CHAIR$_i$ and per-sentence CHAIR$_s$ ($\downarrow$) on COCO2014, with superscript G as the greedy strategy and superscript B as beam search. Additionally, we report the Question Pair Accuracy ($qAcc$), Figure Accuracy ($fAcc$) and All Accuracy ($aAcc$) for Hallusion Bench, along with the GPT score and hallucination rate on MMHAL-Bench. Best results are highlighted in **bold**. Experiments are based on LLaVA-1.5 Vicuna-7B.

| Dataset | POPE | | | COCO2014 | | | | HallusionBench | | |
|---|---|---|---|---|---|---|---|---|---|---|
| Metric | Adv. | Pop. | Random | CHAIR$_s^G$ ($\downarrow$) | CHAIR$_i^G$ ($\downarrow$) | CHAIR$_s^B$ ($\downarrow$) | CHAIR$_i^B$ ($\downarrow$) | $qAcc$ | $fAcc$ | $aAcc$ |
| VIT | 85.17 | 87.30 | **88.47** | 48.60 | 13.40 | 53.40 | 14.90 | 10.33 | 19.36 | 43.85 |
| L2T | **85.60** | **87.90** | **88.47** | **46.20** | **11.80** | **51.40** | **13.10** | **10.77** | 19.36 | **44.90** |

| Dataset | MMHAL-Bench | | | | | | | | | |
|---|---|---|---|---|---|---|---|---|---|---|
| Metric | Score | Hal. Rate ($\downarrow$) | Attribute | Adversarial | Comparison | Counting | Relation | Environment | Holistic | Other |
| VIT | 1.73 | 0.68 | 3.00 | 1.25 | 1.58 | **2.50** | 1.17 | 2.08 | 1.00 | 1.25 |
| L2T | **2.36** | **0.53** | **3.08** | **1.67** | **2.92** | 1.92 | **2.25** | **3.33** | **1.92** | **1.83** |

**Results on POPE.** POPE [Li et al., 2023b] is the most commonly used benchmark for object hallucination. It consists of "Yes or No" questions to evaluate the MLLMs' capability to determine whether the given object is in the image. We report accuracies based on questions derived from adeversarial, popular and random sampling. L2T outperforms VIT in both adversarial and popular sampling settings, showcasing the effectiveness of L2T in reducing object hallucinations.

**Results on CHAIR Evaluation.** In addition to the discriminative evaluations on POPE, we utilize the CHAIR metric [Rohrbach et al., 2018] as a complementary approach to evaluate object hallucination in image captioning tasks. Specifically, CHAIR quantifies the proportion of objects referenced in an image caption that are absent from the corresponding ground-truth label set. As shown in Table 2, our L2T consistently outperforms VIT, achieving improvements of 2.2% and 1.7% in terms of the CHAIR$_s$ and CHAIR$_i$ metric, respectively. This further demonstrates that our L2T effectively regularize MLLMs, reducing the generation of plausible but visually irrelevant contents.

**Results on MMHAL-Bench.** CHAIR mainly focuses on object hallucination by evaluating the presence of objects. To provide a more fine-grained assessment, we further adopt MMHAL-Bench [Sun et al., 2024b] to evaluate a broader range of hallucination types. For evaluating the quality of the generated content, we leverage GPT-4 to compute the GPT score and hallucination rate for comparison. The results demonstrate that L2T significantly outperforms VIT by 36% in average score and 22% in hallucination rate. Moreover, L2T consistently outperforms VIT across various hallucination types, with notable improvements in adversarial, comparison, relation, environment, and holistic settings.

**Results on HallusionBench.** To further improve the comprehensiveness of hallucination evaluation, we use HallusionBench [Guan et al., 2024] to include more scenarios with various disciplines, image types and input modalities. We report Question Pair Accuracy ($qAcc$), Figure Accuracy ($fAcc$) and All Accuracy ($aAcc$) based on GPT-4 for comparison. The results demonstrate that L2T outperforms VIT, with improvements of 0.44% and 1.05% in terms of $qAcc$ and $aAcc$, respectively, highlighting its effectiveness in mitigating a wider range of failure modes in multimodal hallucinations.

## 3.4 Further Analysis

**Mitigate Overfitting from the Loss Perspective.** In Figure 8, we present the loss distributions of our L2T and VIT on the training dataset (VQAv2) and test datasets (DocVQA and NoCaps). The cross-entropy loss is calculated specifically for the response component, excluding the instruction part, even for L2T. The results reveal that L2T exhibits slightly higher loss than VIT on the training data but achieves lower loss on unseen test data. This clearly demonstrates that our method, by learning to instruct images as a regularizer, effectively avoids overfitting on the training data and exhibits superior generalization performance.

**Ablation on the Scale of Instruction Data.** To investigate the impact of different data volumes during the instruction tuning phase on L2T, we present the performance of both the baseline VIT and L2T at 40%, 60%, 80% data levels in Figure 9(a). Note that we use the same scale for the axes of both radar charts. It can be observed that, at the same data scale, L2T significantly outperforms VIT. Specifically, from 40% to 80% instruction data, our method shows an average performance improvement over VIT across the eight metrics in Figure 9(a) by 5.9%, 13.0%, and 9.3%, respectively. The performance

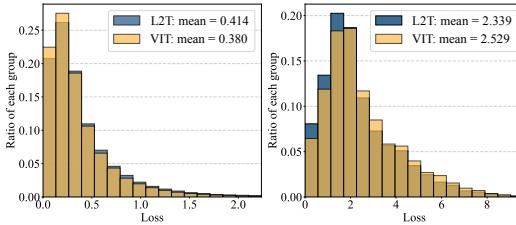

| (a) Train loss on VQAv2. | (b) Test loss on DocVQA. | (a) Instruction data scale. | (b) Pretraining data scale. |

Figure 8: (a) Training loss distribution of L2T and VIT on VQAv2. (b) Testing loss distribution of L2T and VIT on DocVQA. The standard cross-entropy loss is calculated for the response part. Experiments are based on TinyLLaVA Qwen2-0.5B.

Figure 9: Ablation on (a) instruction data scale and (b) pretraining data scale. The figure shows the performance of L2T and VIT using 40%, 60%, 80%, and 100% of the data, with another stage using the full dataset. Experiments are based on TinyLLaVA Qwen2-0.5B.

Table 4: Ablations on template removal. The table shows the performance of VIT, which learns to respond on $X_A$, and L2T, which learns to instruct using variants of progressively removing system template $\mathbf{X}_I^S$ and task template $\mathbf{X}_I^T$. Here, $X_I^{\backslash S,T}$ refers to the instructions excluding both the system and task templates. Experiments are based on TinyLLaVA Qwen2-0.5B.

| Method | | | | Benchmark | | | | | | | | $\Delta$ |
|---|---|---|---|---|---|---|---|---|---|---|---|---|
| $X_I^{\backslash S,T}$ | $\mathbf{X}_I^T$ | $\mathbf{X}_I^S$ | $X_A$ | VQAv2-L | VizWiz-L | MMMU | MMStar | ChartQA | OCR Bench | Flickr30k-L | RefCOCO-L | |
| | | | ✓ | 64.02 | 29.90 | 29.67 | 34.93 | 13.28 | 26.50 | 70.28 | 14.81 | - |
| ✓ | ✓ | ✓ | ✓ | 65.06 | 28.96 | 31.11 | 34.57 | 14.48 | 26.90 | 73.79 | 19.76 | +6% |
| ✓ | ✓ | | ✓ | 65.26 | 29.52 | 31.22 | 35.85 | 13.76 | 27.40 | 74.21 | 21.97 | +9% |
| ✓ | | | ✓ | 66.34 | 29.74 | 32.33 | 36.14 | 13.80 | 27.20 | 74.30 | 23.11 | +11% |

improvements suggest that our method benefits from increasing data volume, highlighting its potential for continuous improvement within the research trend of expanding data scaling laws.

**Ablation on the Scale of Pretraining Data.** We also investigate the impact of data volumes during the pretraining phase on L2T, with the results shown in Figure 9(b). L2T demonstrates a clear advantage over the baseline at the same data scale. From 40% to 80% pretraining data, the average improvements are 10.9%, 8.7%, and 8.5%, respectively. Interestingly, the performance with 80% pretraining data even surpasses that with 100% pretraining data. This "less-is-more" [Zhou et al., 2023] phenomenon highlights the complexity of how pretraining data impacts final performance.

**Ablation on Types of instruction data.** We selected different types of data from the full dataset (mixed with 10% of other types of data), including QA data, GPT-generated data, selection data, grounding data, and captioning data. We conduct instruction tuning using each of these data types, and the results are presented in Table 3. It can be observed that the improvements of our method vary considerably across different data types, which we attribute primarily to the differences in instruction quality. The instructions for grounding data consist only of fixed templates and almost random bounding box coordinates, making it difficult for the model to gain improvements from learning such instructions. In contrast, the instructions for QA data contain a wealth of information related to the image content, which enables our method to provide a significant improvement.

Table 3: Performance comparison of VIT and L2T when using a single data type as the primary training data, including Choice, Grounding, GPT-generated, Captioning, and QA data. Experiments are based on TinyLLaVA Qwen2-0.5B.

| Data | Method | VQAv2-L | MMMU | ChartQA | RefCOCO-L | $\Delta$ |
|---|---|---|---|---|---|---|
| Choice | VIT | 52.76 | 31.56 | 12.56 | 20.45 | - |
| | L2T | 50.90 | 31.22 | 11.88 | 25.33 | +3% |
| Grounding | VIT | 53.10 | 33.76 | 11.32 | 25.01 | - |
| | L2T | 52.62 | 32.11 | 11.32 | 26.51 | +0% |
| GPT-gen | VIT | 53.90 | 31.11 | 11.60 | 8.42 | - |
| | L2T | 55.16 | 31.78 | 11.68 | 11.34 | +10% |
| Captioning | VIT | 45.12 | 29.44 | 11.52 | 16.02 | - |
| | L2T | 47.46 | 31.11 | 12.00 | 16.85 | +5% |
| QA | VIT | 63.26 | 31.00 | 11.84 | 16.94 | - |
| | L2T | 62.28 | 33.00 | 12.36 | 29.69 | +21% |

**Ablation on Template Removal.** To gain more insights, we conduct experiments to assess the impact of removing each prompt template from the instructions, as shown in Table 4. Four experimental setups are considered: (1) the baseline method, which learns only the answer component; (2) the

Table 6: Performance comparison on VLM baseline Prism-7B. L2T yields consistent improvements across diverse benchmarks, highlighting its general effectiveness.

| Task | GQA | VizWiz | TextVQA | RefCOCO | RefCOCO+ | RefCOCOg | POPE | VSR | AI2D |
|------|-----|--------|---------|---------|----------|----------|------|-----|------|
| VIT Prism-7B | 61.92 | 55.36 | 52.80 | 56.70 | 50.70 | 52.70 | 88.00 | 53.20 | 55.50 |
| L2T Prism-7B | **62.62** | **57.75** | **55.60** | **66.00** | **58.90** | **62.00** | **88.50** | **61.70** | **57.10** |

straightforward method, which learns both the full set of instructions and answers; (3) the method that learns the instructions excluding system and format messages (USER/ASSISTANT tokens), while still learning the answers; and (4) the method that learns the instructions without system/format messages and task-specific prompts, while learning the answers as well. The results indicate that progressively removing task-irrelevant template tokens enhances the overall performance of L2T, preventing MLLMs from overfitting to redundant template tokens.

**Computational Analysis.** To validate the computational efficiency of L2T, we conduct experiments on finetuning LLaVA-1.5 Vicuna-7B using instructional data with varying instruction-to-response length ratios ($L_I/L_A$), ranging from 0.05 to 20. The data are sampled from the training set. We report the average number of samples per second and the average number of steps per second (averaged over 100 training steps). Figure 10 demonstrates that L2T achieves an average of $0.331 \pm 0.005$ steps per second, while VIT achieves $0.334 \pm 0.005$. Our L2T incur only a negligible computational overhead of less than 1%. Detailed results can be referred to Table 11 in the Appendix E.

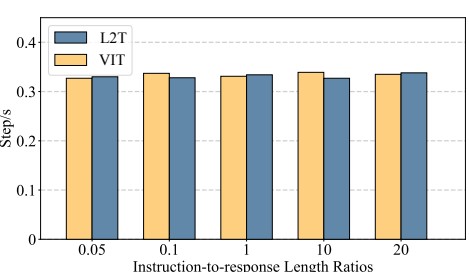

Figure 10: Computational cost across different $L_I/L_A$. Experiments are based on LLaVA-1.5 Vicuna-7B.

**Evaluation on another VLM baseline Prism-7B.** To further verify the generalizability of L2T, we conduct additional experiments on another robust vision-language model (VLM) baseline, Prism [Karamcheti et al., 2024], which surpasses LLaVA-1.5 by incorporating optimized training strategies, advanced image preprocessing, and fused visual backbones such as SigLIP and DiNOv2. We apply L2T to Prism-DINOSigLIP-Controlled-7B and compare it with the standard VIT-style Prism baseline across diverse benchmarks, including visual question answering (GQA, VizWiz, TextVQA), localization (RefCOCO, RefCOCO+, RefCOCOg), and challenging reasoning tasks (POPE, VSR, AI2D). As shown in Table 6, L2T consistently improves performance across all benchmarks, demonstrating its effectiveness and broad applicability.

**Effect on Textual Understanding.**
To examine whether L2T compromises language proficiency by altering the SFT data composition, we evaluate the model on several text-only benchmarks, including MT-Bench, WildBench, MMLU, and AGIEval. We use the LLaVA-1.5 Vicuna-7B

Table 5: Performance comparison on text-only benchmarks. L2T maintains comparable or improved performance, indicating no degradation in core language ability.

| **Benchmark** | MTBench | WildBench | MMLU | AGIEval |
|---------------|---------|-----------|------|---------|
| ViT LLaVA1.5-7B | 5.35 | -9.40 | 49.57 | 32.62 |
| LiT LLaVA1.5-7B | 5.49 | -6.27 | 49.18 | 32.49 |

model as the baseline and apply L2T under identical training configurations. As shown in Table 5, L2T achieves comparable results to the baseline on MT-Bench, MMLU, and AGIEval, while notably improving WildBench performance. These results indicate that learning to instruct functions as a synergistic regularization rather than a disruptive modification, enhancing visual grounding without compromising the model's core text comprehension capabilities.

**Exploring the potential of L2T for self-improving instruction tuning.** To illustrate the broader potential of L2T, we conduct a pilot exploration where the model leverages its own generation capability to enhance itself. Starting from a L2T model pretrained on a 100k subset of the LLaVA-mix-665k dataset, we prompt it with image-only inputs to automatically produce 100k instruction–response pairs. These self-generated samples are then merged with the original training data for continued fine-tuning. As shown in Table 7, this simple self-bootstrapping process leads to consistent gains across multiple benchmarks. The results highlight L2T's potential to evolve through its own generated supervision, paving the way toward self-improving vision–language models.

Table 7: Exploring the potential of L2T for self-improving instruction tuning. Incorporating 100k model-generated instruction–response pairs improves performance across diverse benchmarks.

| Benchmark | VQAv2-L | VizWiz-L | MMMU | MMSTAR | ChartQA | OCR Bench | Flickr30k-L | RefCOCO-L |
|---|---|---|---|---|---|---|---|---|
| L2T TinyLLaVA-0.5B | 56.68 | 23.20 | 31.78 | 33.46 | 11.96 | 23.40 | 68.55 | 17.42 |
| L2T TinyLLaVA-0.5B w/ generated data | **60.80** | **24.22** | **32.33** | **34.75** | **12.44** | **24.50** | 68.32 | **24.53** |

# 4 Related Work

**Visual Instruction Tuning.** Recent advances in multimodal learning [Radford et al., 2021, Zhou et al., 2025, Zhao et al., 2025, Liu et al., 2025] have greatly enhanced the integration of visual and linguistic understanding. Building upon these foundations, the concept of Visual Instruction Tuning was first introduced in LLaVA [Liu et al., 2023a] and MiniGPT-4 [Zhu et al., 2024], aiming to unify the understanding of vision and language by leveraging pre-trained visual and language models. Common system architectures typically consist of (1) a pre-trained visual model for encoding visual features, (2) a pre-trained large language model for interpreting images and user instructions and generating responses, and (3) a cross-modal connector for aligning visual features with the language model's input. Visual resamplers, such as Qformer [Li et al., 2023a], can serve as an optional module to reduce the number of visual patches [Bai et al., 2023a, Dai et al., 2023]. LLaVA-NeXT [Liu et al., 2024b] significantly enhances visual perception by using dynamic visual resolutions. DEEM [Luo et al., 2024] replaces the traditional visual encoder with a diffusion model, further enhancing visual perception. Cambrian-1 [Tong et al., 2024] improves visual robustness through visual encoder routing, but it also introduces higher training overhead.

**(Language) Instruction Tuning.** Instruction tuning has emerged as a critical approach in aligning large language models (LLMs) with specific tasks or domains. By fine-tuning language models on datasets composed of task instructions and corresponding responses, this approach has demonstrated its effectiveness in enhancing generalization to unseen tasks, as evidenced by models like Instruct-GPT [Ouyang et al., 2022] and Flan-PaLM [Chung et al., 2024]. Early explorations of instruction tuning achieved notable success using human-written completions [Bai et al., 2022, Ouyang et al., 2022, Wei et al., 2022]. Recent studies [Wang et al., 2023, Honovich et al., 2023] have expanded on this by exploring how content generated by large language models can be used to construct instruction tuning datasets, further enhancing model capabilities.

The most relevant recent work to ours is IM [Shi et al., 2024], which incorporates loss over instructions during the instruction tuning process of language models. However, our work differs significantly in the following key aspects: 1. Scope: Our work focuses on MLLMs, whereas IM is on language models. 2. Motivation: Our primary motivation is to encourage the model to pay greater attention to visual information and avoid shortcut learning based solely on textual cues. In contrast, IM is primarily aimed at mitigating overfitting. 3. Methodological Differences: We employ an automated approach to filter out the impact of frequently occurring template instructions (see ablation study in Table 4), whereas IM only filters a limited set of special tokens, such as "`<|user|>`". 4. Scalability: We validate the effectiveness of L2T on a large-scale dataset of nearly 1M samples, whereas IM's evaluation is limited to much smaller datasets, showing performance saturation at just 13k samples.

# 5 Conclusion

In this paper, we propose L2T, which enhances multimodal capabilities by learning to instruct images as a regularizer, rather than focusing only on learning to respond. This enables L2T to seamlessly expand the training data, reducing overfitting and proactively guiding the model to learn visual inputs more effectively, thereby avoiding shortcuts. Experimental results demonstrate the effectiveness of L2T across 16 multimodal tasks, highlighting its superior performance on OCR and image captioning tasks by placing greater emphasis on visual content. Furthermore, L2T significantly improves hallucination mitigation. It is also worth noting that our method is orthogonal to existing advancements in MLLMs and can be easily integrated into these methods with minimal compromises on computational costs. We believe that L2T has the potential to evolve the general VIT framework by mitigating overfitting, enhancing data efficiency, and promoting improved generalization in MLLMs.

## Acknowledgement

This work is partially supported by the National Key R&D Program of China (No. 2022ZD0160703), National Natural Science Foundation of China (No. 62306178) and STCSM (No. 22DZ2229005), 111 plan (No. BP0719010). BH is supported by RGC Young Collaborative Research Grant No. C2005-24Y, RGC General Research Fund No. 12200725, and NSFC General Program No. 62376235.

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

# A Detailed Removed Task Templates

In this section, we provide a comprehensive overview of the task templates removed during the training of TinyLLaVA, LLaVA 1.5, and LLaVA-NeXT, as shown in Table 8 and Table 9. Notably, LLaVA-NeXT-Data includes a more diverse set of instruction datasets, leading to a wider variety of task templates. Consequently, a larger number of task templates are removed compared to those discarded from the LLaVA-mix-665k dataset.

Table 8: Detailed removed task templates for training TinyLLaVA, LLaVA 1.5.

| **Task Templates** |
| --- |
| Answer the question using a single word or phrase. |
| Answer with the option's letter from the given choices directly. |
| Provide a one-sentence caption for the provided image. Reference OCR token: |
| Please provide a short description for this region: |
| Please provide the bounding box coordinate of the region this sentence describes: |
| What is the title of this book? |
| What is the genre of this book? |
| What type of book is this? |
| Who is the author of this book? |
| Who wrote this book? |

Table 9: Detailed removed task templates for training LLaVA-NeXT.

| **Task Templates** | |
| --- | --- |
| Answer the question using a single word or phrase. | Answer the question with GPT-T-COCO format. |
| Provide a short description for the given region. | What is the title of this book? |
| Answer with the option's letter from the given choices directly. | Provide the bounding box coordinates of the region that the given sentence describes. |
| OCR this image section by section, from top to bottom, and left to right. | Do not insert line breaks in the output text. |
| If a word is split due to a line break in the image, use a space instead. | What type of book is this? |
| Who is the author of this book? | Describe this image in detail with GPT-T-COCO format. |
| Provide a one-sentence caption for the provided image. | Provide the requested information directly. |
| What is the genre of this book? | Answer the question with a single word. |
| Are the values in the chart presented in a percentage scale? | Is each bar a single solid color without patterns? |
| Which group has the smallest summed value? | Which group has the largest summed value? |
| Is this book related to? | Does the chart contain stacked bars? |
| What is the label of the second bar from the left in each group? | What is the label of the first bar from the left in each group? |
| Which group of bars contains the smallest valued individual bar in the whole chart? | What is the value of the smallest individual bar in the whole chart? |
| Which group of bars contains the largest valued individual bar in the whole chart? | What is the value of the largest individual bar in the whole chart? |
| How many bars are there? | Which bar has the largest value? |
| What is the value of the largest bar? | Which bar has the smallest value? |
| Does the chart contain any negative values? | Are the values in the chart presented in a logarithmic scale? |
| Which algorithm has the smallest accuracy summed across all the datasets? | Which algorithm has the largest accuracy summed across all the datasets? |
| What is the label of the second group of bars from the left? | Which object is preferred by the least number of people summed across all the categories? |
| Which object is preferred by the most number of people summed across all the categories? | What is the setting of the image? |
| What is the label of the first group of bars from the left? | Which item sold the least number of units summed across all the stores? |
| Which object is the most preferred? | What is the label of the first bar from the left? |
| Which item sold the most number of units summed across all the stores? | How many bars are there per group? |
| What is the label of the second bar from the left? | Which object is the least preferred? |
| How many groups of bars are there? | Which object is the least preferred in any category? |
| Which algorithm has the lowest accuracy? | What is the accuracy of the algorithm with lowest accuracy? |
| Which algorithm has the highest accuracy? | What is the accuracy of the algorithm with highest accuracy? |
| Which algorithm has highest accuracy for any dataset? | What is the highest accuracy reported in the whole chart? |
| Which algorithm has lowest accuracy for any dataset? | What is the lowest accuracy reported in the whole chart? |
| Which object is the most preferred in any category? | What is the label of the third group of bars from the left? |
| What is the label of the second bar from the bottom in each group? | What is the label of the first bar from the bottom in each group? |
| What is the difference between most and least preferred object? | What is the difference between the largest and the smallest value in the chart? |
| How much more accurate is the most accurate algorithm compared to the least accurate algorithm? | What is the label of the third bar from the left? |
| Which item sold the least units? | How many units of the least sold item were sold? |
| How many people prefer the most preferred object? | Which item sold the most units? |
| How many units of the most sold item were sold? | Which item sold the most units in any shop? |
| How many units did the best selling item sell in the whole chart? | How many people prefer the least preferred object? |
| Which item sold the least units in any shop? | How many units did the worst selling item sell in the whole chart? |
| How many people like the least preferred object in the whole chart? | How many people like the most preferred object in the whole chart? |
| What is the label of the third bar from the left in each group? | What is the label of the fourth group of bars from the left? |
| What is the label of the first group of bars from the bottom? | How many more of the most sold item were sold compared to the least sold item? |
| What is the label of the second group of bars from the bottom? | What is the label of the second bar from the bottom? |
| What is the label of the first bar from the bottom? | What is the label of the third group of bars from the bottom? |

# B More Details on Experimental Setup

## B.1 More Details on Hyperparameter

Detailed hyperparameters for training TinyLLaVA, LLaVA 1.5 and LLaVA-NeXT are shown in Table 10. For TinyLLaVA and LLaVA 1.5, pretraining is conducted with a learning rate of 1e-3 and a batch size of 256, while finetuning uses a learning rate of 2e-5 and a batch size of 128. For LLaVA-NeXT, pretraining uses a learning rate of 1e-3 and a batch size of 128, with finetuning using

a learning rate of 1e-5 and a batch size of 32. All experiments use the AdamW optimizer [Loshchilov and Hutter, 2019] and a cosine decay schedule with a warm up ratio of 0.03. During pretraining phase, only the multimodal connector is trained. In finetuning, TinyLLaVA and LLaVA 1.5 jointly train the connector and language model, while LLaVA-NeXT trains all parameters, including the vision encoder.

Table 10: Training Hyperparameters for TinyLLaVA, LLaVA 1.5 and LLaVA-NeXT.

| Hyperparameter | TinyLLaVA & LLaVA 1.5 | | LLaVA-NeXT | |
| --- | --- | --- | --- | --- |
| | Pretrain | Finetune | Pretrain | Finetune |
| Learning rate (LR) | 1e-3 | 2e-5 | 1e-3 | 1e-5 |
| LR warmup ratio | 0.03 | 0.03 | 0.03 | 0.03 |
| Batch size | 256 | 128 | 128 | 32 |
| LR schedule | cosine decay | cosine decay | cosine decay | cosine decay |
| Epoch | 1 | 1 | 1 | 1 |
| Optimizer | AdamW | AdamW | AdamW | AdamW |
| Trainable parameters | MLP | MLP, LLM | MLP | Vision enc., MLP, LLM |

## B.2 More Details on Training Data

In the finetuning stage of instruction tuning, we use the LLaVA-mix-665k data [Liu et al., 2024a] on TinyLLaVA and LLaVA 1.5, which incorporates a diverse set of instruction-following datasets, including free conversational data (LLaVA-Instruct [Liu et al., 2023a]), visual question answering (VQAv2 [Goyal et al., 2017], GQA [Hudson and Manning, 2019], OKVQA [Marino et al., 2019], A-OKVQA [Schwenk et al., 2022]), OCR and captioning (OCRVQA [Mishra et al., 2019], TextCaps [Sidorov et al., 2020]), and visual grounding (RefCOCO [Kazemzadeh et al., 2014], VG [Krishna et al., 2017]). For LLaVA-NeXT, we use the LLaVA-NeXT-Data [Liu et al., 2024b], which extends the LLaVA-mix-665k [Liu et al., 2024a] dataset with high-quality user instruction data from LAION-GPT-V, ShareGPT-4V [Chen et al., 2024a], and LLaVA-demo [Liu et al., 2023a], as well as OCR, document, and chart data from DocVQA [Mathew et al., 2021], SynDog-EN [Kim et al., 2022], ChartQA [Masry et al., 2022], DVQA [Kafle et al., 2018], and AI2D [Kembhavi et al., 2016].

## C More Details on CHAIR Evaluation

As mention in Section 3.3, we utilize the CHAIR metric to evaluate object hallucination in image captioning tasks. The metric assesses object hallucinations through two dimensions: per-instance $CHAIR_i$ and per-sentence $CHAIR_s$. The former measures the fraction of object instances that are hallucinated, while the latter determines the proportion of sentences that contain at least one hallucinated object. Definitions of these two metrics are formally provided:

$$CHAIR_i = \frac{|\{\text{hallucinated objects}\}|}{|\{\text{all objects mentioned}\}|},$$

$$CHAIR_s = \frac{|\{\text{sentences with hallucinated object}\}|}{|\{\text{ all sentences}\}|}.$$

We follow [Huang et al., 2023] to perform the CHAIR evaluation on the COCO2014 dataset. Specifically, we randomly select 500 images in the validation set. The decoding process uses the greedy strategy and the Beam search with $N_{beam} = 5$, with a maximum of 512 tokens for new content generated.

## D More Cases

More cases of our L2T compared to VIT are shown in Figure 11. Our L2T consistently outperforms VIT across a range of scenarios, including OCR, image captioning, and hallucination mitigation.

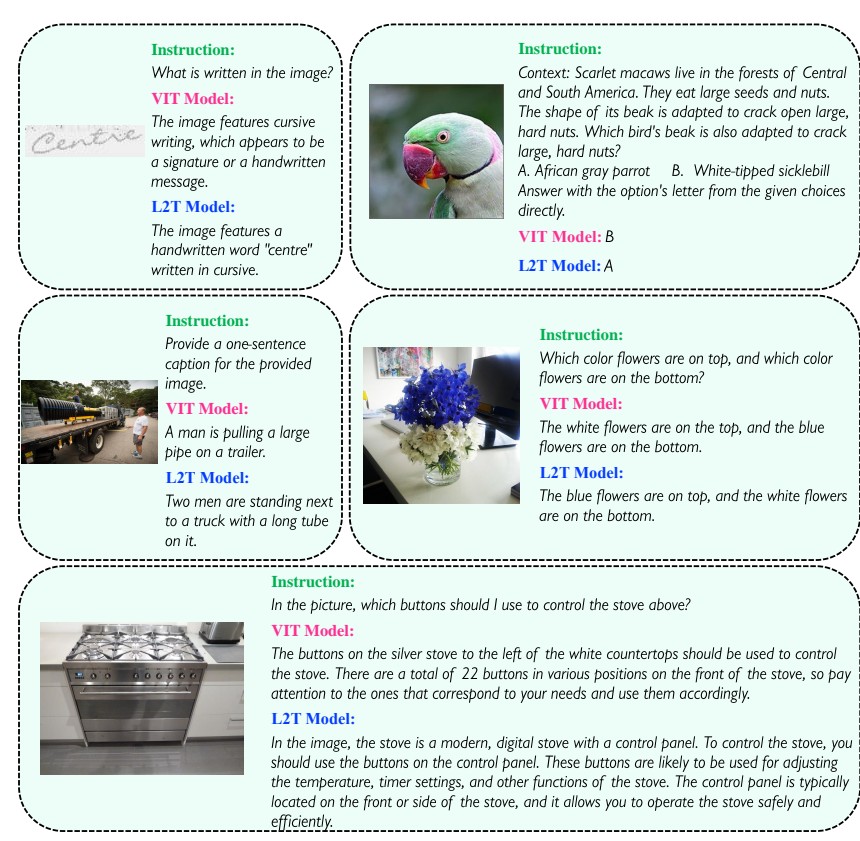

Figure 11: More cases of L2T and VIT across OCR, image captioning, and hallucination mitigation tasks.

# E Computational Cost

Table 11 presents the computational cost of VIT and L2T across different instruction-to-response length ratios. The experiments are conducted on 8 NVIDIA A100 GPUs.

Table 11: Computational cost of VIT and L2T across different instruction-to-response length ratios. We report the number of samples per batch and steps per batch. Experiments are based on LLaVA-1.5 Vicuna-7B.

| Q/A Ratio | 0.05 | | 0.1 | | 1 | | 10 | | 20 | |
|---|---|---|---|---|---|---|---|---|---|---|
| Metric | sample/s | step/s | sample/s | step/s | sample/s | step/s | sample/s | step/s | sample/s | step/s |
| VIT | 2.617 | 0.327 | 2.692 | 0.337 | 2.645 | 0.331 | 2.708 | 0.339 | 2.678 | 0.335 |
| L2T | 2.640 | 0.330 | 2.623 | 0.328 | 2.676 | 0.334 | 2.613 | 0.327 | 2.708 | 0.338 |

# F Visualization of Attention Heads

Recent observations [Orgad et al., 2025, Shen et al., 2025] show that the hidden activation of the token preceding the answer (the colon ":", in the prompt "ASSISTANT:") encodes more information than the output logits. Inspired by this, we select this token's hidden activation for visualization. Specifically, we visualize 14 attention heads from the last layer of a 24-layer transformer during inference, focusing on their activation across input tokens, particularly image tokens, to compare VIT and L2T. Data samples are randomly selected from VQAv2 [Goyal et al., 2017], GQA [Hudson and Manning, 2019] and OKVQA [Marino et al., 2019]. Detailed information about the selected examples is provided in Table 12, and the visualizations of the corresponding attention heads are presented in Figure 6 and Figure 12.

Table 12: Examples for visualization of attention heads.

| Prompt | Image | Visualization |
|---|---|---|
| A chat between a curious user and an artificial intelligence assistant. The assistant gives helpful, detailed, and polite answers to the user's questions. USER: <image>\nOn which side of the photo is the faucet, the right or the left?\nAnswer the question using a single word or phrase. ASSISTANT: | 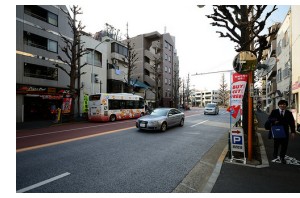 | Figure 6 |
| A chat between a curious user and an artificial intelligence assistant. The assistant gives helpful, detailed, and polite answers to the user's questions. USER: <image>\nIs the car to the left or to the right of the man?\nAnswer the question using a single word or phrase. ASSISTANT: | 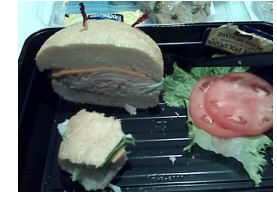 | Figures 12(a) and 12(b) |
| A chat between a curious user and an artificial intelligence assistant. The assistant gives helpful, detailed, and polite answers to the user's questions. USER: <image>\nWhere these red vegetables imported to the us or exported from the us?\nAnswer the question using a single word or phrase. ASSISTANT: | 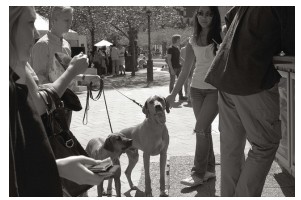 | Figures 12(c) and 12(d) |
| A chat between a curious user and an artificial intelligence assistant. The assistant gives helpful, detailed, and polite answers to the user's questions. USER: <image>\nIn which knee is the hole in the women's pants?\nAnswer the question using a single word or phrase. ASSISTANT: | | Figures 12(e) and 12(f) |

# G Discussion on the Scope of This Work

Recent vision–language models (VLMs) often struggle with challenges such as hallucination and shortcut learning, which can limit their reliability and generalization. Addressing these issues requires careful consideration of the interactions between different training stages, including Supervised Fine-Tuning (SFT), rule-based Reinforcement Learning (RL), and Reinforcement Learning from Human Feedback (RLHF).

Our work focuses on improving the foundational SFT stage. We posit that the quality of early instruction tuning directly influences the effectiveness and safety of subsequent alignment. Indeed, recent studies indicate that applying RL or RLHF on models already prone to hallucination or shortcut learning can reinforce these issues, leading to confidently incorrect outputs.

L2T addresses this problem at its root by reducing shortcut learning and enhancing visual grounding during SFT. A stronger SFT foundation provides a more reliable base for RLHF, enabling safer and more effective alignment. While recent work has emphasized RL and reward modeling as post-training interventions, foundational flaws in SFT can persist or even intensify during alignment. Thus, improving early-stage instruction tuning serves as a preventive measure rather than relying solely on corrective post-training methods.

Overall, this perspective highlights the broader impact of L2T: by enhancing the base model during SFT, it facilitates downstream alignment and helps define the scope of our contribution in enabling safer and more generalizable multimodal training strategies.

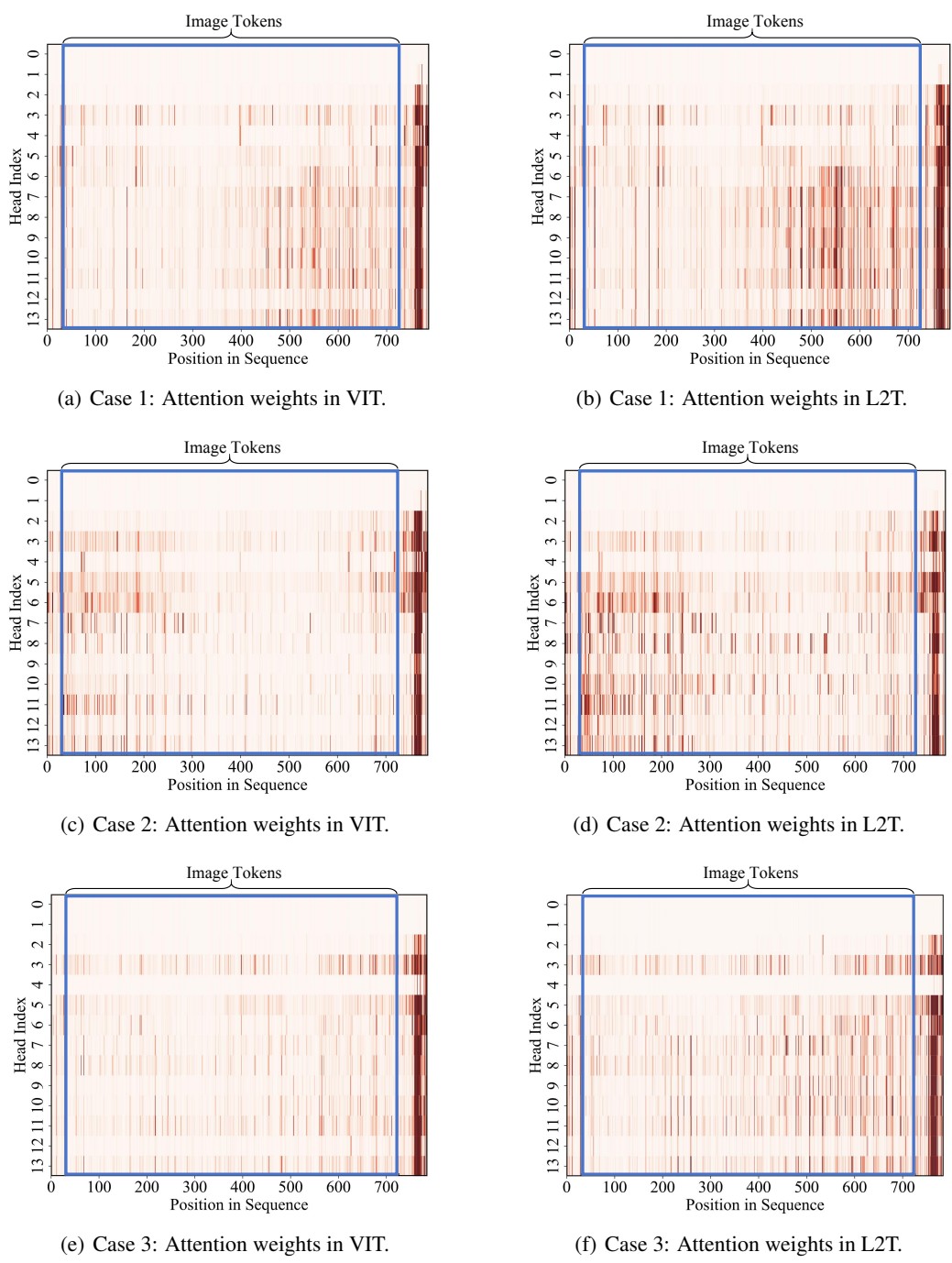

(a) Case 1: Attention weights in VIT.

(b) Case 1: Attention weights in L2T.

(c) Case 2: Attention weights in VIT.

(d) Case 2: Attention weights in L2T.

(e) Case 3: Attention weights in VIT.

(f) Case 3: Attention weights in L2T.

Figure 12: Visualization of attention weights in VIT and L2T for additional cases. Darker colors indicate higher attention weights. Experiments are based on TinyLLaVA Qwen2-0.5B.

## H  Limitations

Despite the promising capabilities underscored by L2T, several limitations must be acknowledged. First, the performance improvements achieved by L2T mainly stem from the informative instruction content that aids in better understanding visual inputs. In contrast, redundant instruction content, such as system or task templates, do not contribute to performance gains, as they lack relevant information about the visual content. Second, it is critical to ensure that the instructions do not contain harmful or

biased content. The presence of such content could raise concerns about the fairness and reliability of MLLMs, making them more susceptible to generating flawed, biased, or even harmful responses.

# I   Broader Impact

This work focuses on improving vision instruction tuning techniques, advancing the multimodal capabilities. It can benefit applications such as assistive technologies, education, and content creation. For example, individuals with visual impairments could gain access to enhanced image-to-text descriptions, improving accessibility and inclusivity. However, alongside the development of multimodal technologies, it is crucial to address the risks of bias and harmful content. The model may inadvertently propagate societal biases embedded in the training data, leading to discriminatory or inappropriate outputs. Furthermore, enhanced capabilities could be exploited to generate convincing but harmful misinformation or deepfakes, posing risks to societal trust and safety. To responsibly advance this technology, we emphasize the importance of robust mitigation strategies, including diverse and inclusive training data, bias detection tools, and ethical safeguards against misuse. Ensuring that these technologies serve the public good requires ongoing collaboration among researchers, developers, and policymakers, fostering innovation while minimizing unintended negative impacts.

