# OpenReview forum: "Learning to Instruct for Visual Instruction Tuning"
_NeurIPS.cc/2025/Conference — NeurIPS 2025 poster_

### Official Review · Reviewer_xAmq · 2025-06-22

**Clarity:** 3
**Significance:** 3
**Originality:** 3
**Rating:** 5
**Confidence:** 4

**Summary:**

The paper proposes a simple yet effective method to improve visual instruction tuning by incorporating the loss function into both the instruction and response sequences. This approach addresses the overfitting and shortcut learning issues commonly found in Visual Instruction Tuning (VIT). The authors conduct extensive experiments across 16 different multimodal benchmarks. The results demonstrate that LIT outperforms VIT in most cases, with an average relative improvement of up to 8.5%. This comprehensive evaluation provides strong evidence of the effectiveness of LIT.

**Questions:**

Please refer to [Weaknesses].

**Ethical Concerns:**

["NO or VERY MINOR ethics concerns only"]

**Final Justification:**

It is an interesting paper. My corcerns are well-sovled.

**Limitations:**

Please refer to [Weaknesses].

**Quality:**

3

**Strengths And Weaknesses:**

[Strengths]
* The shortcut learning issues are important in visual instruction tuning. This paper points out that masking the instruction in visual instruction tuning is not correct. This solution is interesting and has been ignored by the community.
* The proposed method is simple and effective.

[Weaknesses]
* The process of removing task templates in LIT is somewhat ad-hoc. The authors identify task templates by calculating the frequency of sentences in the training dataset and selecting the most frequent ones. However, this approach may not be robust across different datasets or tasks. A more systematic and adaptive method for template removal could enhance the generalizability of LIT.
* This solution is effective in VQA. As discussed in SP, for the task with a fixed template, this method does not work. Can we design some methods to select which task should not mask the instruction during loss calculation?
* After the training, the model can generate the instructions for a given image. It is interesting to make some generated instructions. Will these instructions be used as training data to further train the model?

---

> ### Author Rebuttal · Authors · 2025-07-31
>
> We thank the reviewer for the thoughtful comments and for recognizing the importance of addressing shortcut learning in visual instruction tuning. We appreciate your positive feedback on the simplicity and effectiveness of our method. Below, we respond to your suggestions in detail.
>
> > **Q1:** The process of removing task templates in LIT is somewhat ad-hoc. The authors identify task templates by calculating the frequency of sentences in the training dataset and selecting the most frequent ones. However, this approach may not be robust across different datasets or tasks. A more systematic and adaptive method for template removal could enhance the generalizability of LIT.
>
> **A1:** We appreciate the reviewer’s insightful questions. Following your valuable suggestion, we conducted an adaptive method using visual contribution (VC) at a more fine-grained token level, as detailed in Section 2.  Specifically, we computed the VC for each token in the instruction part, defined as:
>
> $\mathrm{VC}\^\mathrm{I}\_i = \log p\_{\theta} (\mathbf{X}\_{I,i}|\mathbf{X}\_V,\mathbf{X}\_{I,<i}) - \log p\_{\theta} (\mathbf{X}\_{I,i}|\mathbf{X}\_V=\emptyset,\mathbf{X}\_{I,<i}).$
>
> This VC score allows us to identify the least important tokens with the lowest visual contribution across the dataset. By setting a threshold of 20%, we filtered out the most irrelevant tokens, intuitively removing task templates that likely conveyed minimal information related to the images. For experimental details, we first pretrained a TinyLLaVA Qwen 0.5B model to estimate token-level VC. The resulting VC scores were then used to train a new model from scratch, during which meaningless tokens were removed based on their low VC values.
>
> | Benchmark                     | VQAv2-L | VizWiz-L | MMMU  | MMSTAR | ChartQA | OCR Bench | Flickr30k-L | RefCOCO-L |
> |------------------------------|------------|-------------|-------|--------|---------|------------|----------------|---------------|
> | ViT TinyLLaVA-0.5B           | 64.02      | 29.90       | 29.67 | 34.93  | 13.28   | 26.50      | 70.28          | 14.81         |
> | LiT TinyLLaVA-0.5B w/ VC removal | 64.46  | 30.42       | 31.00 | 36.42  | 13.88   | 27.10      | 74.15          | 20.42         |
> | LiT TinyLLaVA-0.5B           | 66.34      | 29.74       | 32.33 | 36.14  | 13.80   | 27.20      | 74.30          | 23.11         |
>
>
> The results revealed that the VC-based removal did not consistently outperform our template removal method. Additionally, it introduces significant computational overhead, as it requires pretraining and inference on the entire dataset. In comparison, our proposed template removal remains more efficient and effective. However, we agree that the frequency-based removal approach may lack robustness across different datasets or tasks and will consider more adaptive and efficient template removal strategies in future work. We will incorporate this point to the conclusion of the revised manuscript.
>
>
> > **Q2:** This solution is effective in VQA. As discussed in SP, for the task with a fixed template, this method does not work. Can we design some methods to select which task should not mask the instruction during loss calculation?
>
> **A2:** We sincerely appreciate the reviewer for raising this thoughtful question. We would like to clarify as follows:
> - While the performance gains varies across tasks, we would like to clarify that the application of LIT does not lead to performance degradation or incur additional computational overhead. Even for tasks with modest gains, LIT maintains performance comparable to the baseline. Therefore, selectively applying LIT may not be necessary. A unified application ensures simplicity, generality, and no additional computational cost, without compromising the model’s overall performance.
> - The tasks where our method shows relatively limited gain tend to have a key characteristic: their instructions are mostly composed of fixed, uninformative templates. These templates are effectively removed through the template removal process in LiT. For example, in grounding tasks, an instruction such as "Please provide the bounding box coordinate of the region this sentence describes: a red jeep" is reduced to just "a red jeep." Therefore, there is no need to design an additional mechanism to selectively avoid learning from such tasks because the template removal process already handles this. In fact, the limited improvement of LiT on specific tasks is due to the removal of numerous templates in the instructions. The underlying mechanism is that the remaining few tokens convey insufficient information, making it challenging to learn effective instructions.
>
> We will incorporate the clarifications and discussion into our revised manuscript.
>
> > **Q3:** After the training, the model can generate the instructions for a given image. It is interesting to make some generated instructions. Will these instructions be used as training data to further train the model?
>
> **A3:** Thank you for the excellent question. It indeed opens up a promising direction for extending the capabilities of our LiT method. In response to your suggestion, we conducted an experiment to evaluate the feasibility of using model-generated instructions as additional training data for further improving the model. Specifically, we first pretrained the LiT model on a randomly selected 100k subset of the LLaVA‑mix‑665k dataset. Then, we used image-only inputs to guide the model to generate 100k instruction-response pairs. These synthesized samples were then integrated with the original 100k training samples for further training.
>
> | Benchmark                        | VQAv2-L | VizWiz-L | MMMU  | MMSTAR | ChartQA | OCR Bench | Flickr30k-L | RefCOCO-L |
> |----------------------------------|---------|----------|-------|--------|---------|------------|--------------|------------|
> | LiT TinyLLaVA-0.5B               | 56.68   | 23.20    | 31.78 | 33.46  | 11.96   | 23.40      | 68.55        | 17.42      |
> | LiT TinyLLaVA-0.5B w/ generated data | 60.80   | 24.22    | 32.33 | 34.75  | 12.44   | 24.50      | 68.32        | 24.53      |
>
>
> The experimental results demonstrate that incorporating the additional 100k model-generated instruction-response pairs into further training leads to noticeable performance gains. This finding highlights the potential of LiT to improve through self-generated supervision, suggesting a promising direction toward developing self-evolving vision-language models via a closed-loop instruction–response mechanism.
>
> We will include this experiment in the revised manuscript and explore this further in a systematic way in future work. We thank the reviewer again for the constructive idea.

---

> > ### Comment · Reviewer_xAmq · 2025-08-01
> >
> > Thanks for the reply. My concerns are solved. The response to Q3 is quite interesting and surprising. I highly recommend that the authors revise their paper carefully in the final version.

---

> > > ### Author Response · Authors · 2025-08-01
> > >
> > > Thank you for your insightful comment and excellent question on model-generated instructions. We sincerely appreciate your suggestion, and we will incorporate this idea and the corresponding experiments in the revision. We will also include all other discussions and experiments. Thank you again for your dedicated and valuable contribution in reviewing our paper!

---

### Official Review · Reviewer_Qwwm · 2025-07-01

**Clarity:** 3
**Significance:** 3
**Originality:** 2
**Rating:** 4
**Confidence:** 4

**Summary:**

This paper introduces a simple-yet-effective instruction tuning method named LIT: by adding auxiliary training objectives during the SFT (Supervised Fine-Tuning) process, the model learns how to ask questions during training, rather than just learning to answer them. Ultimately, results show that this method can improve the model performance based on llava-series model.

**Questions:**

Please refer to the Strengths and Weaknesses.

**Ethical Concerns:**

["NO or VERY MINOR ethics concerns only"]

**Final Justification:**

Thank the author for the response. I think the response to Q1 could better help the author understand this work. Overall, I tend to maintain the score (rating 4), and I'm happy to see this submission appearing in the conference for the community.

**Limitations:**

Please refer to the Strengths and Weaknesses.

**Paper Formatting Concerns:**

No formatting concerns.

**Quality:**

3

**Strengths And Weaknesses:**

The entire article is coherent and clear, with the method being concise and effective. The following are some issues that require further clarification:

1. This seems to be a general method. Apart from the LLava series, have you considered validating its effectiveness on other models?

2. Is there any source or basis for the design of this LIT approach?

3. This method appears somewhat similar to multimodal pretraining. What are the main design between the two?

---

> ### Author Rebuttal · Authors · 2025-07-31
>
> We thank the reviewer for the positive and clear feedback. We appreciate your recognition of the method’s simplicity and effectiveness. Below, we address your questions regarding model generalization, the design motivation behind LIT, and its relation to multimodal pretraining.
>
> > **Q1:** This seems to be a general method. Apart from the LLava series, have you considered validating its effectiveness on other models?
>
> **A1:** We sincerely thank the reviewer for recognizing the general applicability of our method. Following your valuable suggestion, we have conducted additional experiments on Prism-7B [1], another powerful MLLM in addition to the LLaVA series. We conducted experiments based on Prism-DINOSigLIP-Controlled-7B, applying LiT during training and comparing its performance to the standard VIT-style baseline across several benchmarks, including VQA (GQA, VizWiz, TextVQA), Localization (RefCOCO, RefCOCO+, RefCOCOg), and challenge tasks (POPE, VSR, AI2D).
>
>
> | Task         | GQA   | VizWiz | TextVQA | RefCOCO | RefCOCO+ | RefCOCOg | POPE  | VSR   | AI2D  |
> |--------------|-------|--------|---------|---------|----------|----------|-------|-------|-------|
> | ViT Prism-7B | 61.92 | 55.36  | 52.80   | 56.70   | 50.70    | 52.70    | 88.00 | 53.20 | 55.50 |
> | LiT Prism-7B | 62.62 | 57.75  | 55.60   | 66.00   | 58.90    | 62.00    | 88.50 | 61.70 | 57.10 |
>
> As the results demonstrate, LIT brings consistent and significant performance gains to Prism across a wide range of capabilities. This further substantiates our claim that LIT is a general and widely applicable method for improving visual instruction tuning, with benefits that are not confined to a specific model family. We will include the experiment and discussion in our revised manuscript.
>
> > **Q2:** Is there any source or basis for the design of this LIT approach?
>
> **A2:** Thank you for the question. The design of the LIT method is primarily motivated by direct observations of two prevalent issues in existing models: overfitting and shortcut learning.
>
> Conceptually, LIT draws some inspiration from two well-established ideas in the machine learning literature:
>
> - Auxiliary Tasks in Self-Supervised Learning [2,3]: We treat "instruction generation" as an auxiliary task alongside the primary task of "answer generation". This auxiliary objective compels the model to develop a deeper understanding of the image content, acting as an effective regularizer that enhances the model’s generalization ability.
> - Reward Hacking in Reinforcement Learning [4,5]: During training, models often exploit language priors to "guess" answers while ignoring the visual input, which is an issue closely resembling the "reward hacking" phenomenon in reinforcement learning. LIT addresses this by restructuring the learning objective, requiring the model to generate image-grounded instructions. This effectively blocks the shortcut and ensures the model acquires genuine cross-modal understanding.
>
> Therefore, LIT offers a principled and practical solution that mitigates shortcut learning and overfitting, ultimately leading to more robust and faithful vision-language reasoning. We will incorporate the discussion into our revised manuscript.
>
> > **Q3:** This method appears somewhat similar to multimodal pretraining. What are the main design between the two?
>
> **A3:** Thank you for the excellent question, which allows us to clarify this important distinction. While both processes involve learning from image-text data, our Learning to Instruct (LIT) method is fundamentally different from multimodal pre-training in its stage, objective, and methodology.
>
> In essence, pre-training teaches the model a general association between images and text, while LIT is a specific regularization technique applied during the fine-tuning stage to improve the model's ability to follow instructions and reason about visual content.
>
>
>
> | Aspect         | Multimodal Pre-training                                                                                                        | Learning to Instruct (LIT)                                                                                                                                           |
> | -------------- | ------------------------------------------------------------------------------------------------------------------------------ | -------------------------------------------------------------------------------------------------------------------------------------------------------------------- |
> | Training Stage | The initial phase designed to align general visual features with the language model's embedding space.                         | A technique applied exclusively during the subsequent fine-tuning stage, where the model learns to follow specific instructions.                                     |
> | Objective      | General vision-language feature alignment. The goal is to train a connector that maps visual features into the language space. | Regularization to improve instruction following and reduce shortcut learning. It forces the model to focus on visual content rather than relying on language priors. |
> | Training Data  | Typically uses large-scale, descriptive image-caption pairs.                                                                   | Operates on curated instruction-response datasets (e.g., VQA, OCR, reasoning data).                                                                                   |
> |      Methodology          |       Training only a cross-modal connector while keeping the vision and language models frozen.                                     |                                                                             Extends the standard fine-tuning loss to include the task of generating the instruction itself, conditioned on the image.                                                                                          |
>
> In summary, LIT is not a pre-training method but a novel fine-tuning technique designed to regularize the model, enhance its visual grounding, and mitigate issues like hallucination that arise during the instruction-tuning process. We will incorporate the clarifications and discussion into our revised manuscript.
>
>
> [1] Prismatic VLMs: Investigating the Design Space of Visually-Conditioned Language Models. ICML 2024.
>
> [2] Survey on self-supervised learning: Auxiliary pretext tasks and contrastive learning methods in imaging. Entropy 2022.
>
> [3] Self-Supervised Generalisation with Meta Auxiliary Learning. NeurIPS 2019.
>
> [4] Concrete Problems in AI Safety. 2016.
>
> [5] Reward tampering problems and solutions in reinforcement learning: A causal influence diagram perspective. Synthese 2021.

---

> > ### Comment · Reviewer_Qwwm · 2025-08-04
> >
> > Thank the author for the response. I think the response to Q1 could better help the author understand this work. Overall, I tend to maintain the score, and I'm happy to see this submission appearing in the conference for the community.

---

> > > ### Author Response · Authors · 2025-08-04
> > >
> > > Thank you for your thoughtful review and constructive feedback. We will include Q1 to better convey the method's generality and will incorporate all added discussions and experiments into the revision. We sincerely appreciate your dedicated and valuable contribution in improving the paper!

---

### Official Review · Reviewer_S2mP · 2025-07-02

**Clarity:** 3
**Significance:** 3
**Originality:** 3
**Rating:** 5
**Confidence:** 5

**Summary:**

* The paper focuses on a well-known limitation of Visual Instruction Tuning — that it can lead to “shortcut learning,” where models come to rely on language priors instead of the visual input.
* To address this problem, it proposes Learning to Instruct (LIT), a simple but effective modification to the instruction tuning training objective.
  * → Instead of only calculating loss on the model's *response*, LIT also includes the loss from predicting the *instruction* itself, conditioned on the image.
  * This acts as a regularizer, forcing the model to ground its understanding more deeply in the visual content.
* The method requires no extra data or significant compute overhead and demonstrates consistent performance gains across several models and 16 benchmarks, with particularly strong improvements in OCR & Chart and Image Captioning.

**Questions:**

* **Controlling \# tokens in the loss calculation**: The proposed LIT method computes the loss over more tokens (instruction \+ response) than the VIT baseline (response only). This raises a question about a potential confounding variable: *could the observed benefits be partially attributed to simply having more tokens contributing to the gradient per sample, rather than the specific regularizing effect of predicting the instruction's content?* Have you considered or can you suggest an experiment to control for this and help disentangle these two potential effects?
  * Perhaps this can be accomplished by using the same number of tokens from the system prompt, which should be uninformative.
* **Method Name:** The acronym "LiT" is already widely recognized in the community for "Locked-image Tuning" ([https://arxiv.org/abs/2111.07991](https://arxiv.org/abs/2111.07991)). I am concerned that the current “LIT” name will get overshadowed by the well-established LiT method. Maybe it is worth considering an alternative name?
* **Visualizations & Clarity:** please see questions in the Weaknesses section.

**Ethical Concerns:**

["NO or VERY MINOR ethics concerns only"]

**Final Justification:**

all concerns addressed. this is a strong paper worthy of acceptance.

**Limitations:**

yes

**Paper Formatting Concerns:**

No major concerns. See weaknesses for minor concerns / presentation suggestions.

**Quality:**

4

**Strengths And Weaknesses:**

**Strengths:**

* The paper addresses the important problem of shortcut learning with a clever, simple, and highly practical method. Including the instruction in the loss is a relatively simple modification to Visual Instruction Tuning that could be easily adopted by the community with negligible overhead.
* The LIT method shows consistent performance improvements over strong VIT baselines across multiple model scales.
  * Gains are particularly impressive on vision-centric tasks like OCR, chart understanding, and especially image captioning, where it improves CIDEr by nearly 18% on average.
* **Novel Analysis of Shortcut Learning:** The "Visual Contribution" metric introduced is intuitive and simple to compute, and provides a rigorous way to quantify the model's reliance on visual information. This experiment strengthens the paper's core claim that LIT improves visual grounding.
  * → I am unaware of previous similar analyses. Most papers rely on the *overall* accuracy difference between “blind” and normal (vision-enabled) models on a dataset, which gives a single dataset-level number. The Visual Contribution enables a more precise per-example measure.
* Presentation of the method is clear, Figs. 3 & 4 clearly convey the method.

**Weaknesses:**

* **Poor Data Visualization:** The paper's clarity is significantly hampered by poor choices in data visualization, which make the results difficult to interpret.
  * The primary results in Figs. 1 & 9 are presented as radar charts, which are known to be problematic for making clear comparisons between series (see [this](https://www.darkhorseanalytics.com/blog/radar-more-evil-than-pie) and [this](https://www.data-to-viz.com/caveat/spider.html)).
    * Fig. 9 is especially bad — it is impossible to discern any clear takeaway from the chart.
    * Redoing these visualizations in another format such as a line or bar chart could really help readers glean the message more readily.
  * Fig. 10: the ylims on this chart make no sense. The average is reported to 3 decimals in the paper, but the chart ylims range is 400x larger. The differences between the series is almost undistinguishable from the chart. Maybe consider ylims such as (0.3, 0.35) to zoom in on the relevant portion?
* **Minor Clarity Issues:**
  * Some sentences in the paper are difficult to parse and could be improved with grammar checking and rephrasing for better readability (e.g., L52-54).
  * Fig. 11: what specific models generated these? Should specify in the caption?

---

> ### Author Rebuttal · Authors · 2025-07-31
>
> We sincerely thank the reviewer for the encouraging and constructive feedback. We appreciate your recognition of our method’s simplicity, effectiveness, and novel analysis. Below, we address your suggestions in detail.
>
> > **Q1:** Poor Data Visualization: The paper's clarity is significantly hampered by poor choices in data visualization, which make the results difficult to interpret.
> > The primary results in Figs. 1 & 9 are presented as radar charts, which are known to be problematic for making clear comparisons between series (see this and this).
> > (1) Fig. 9 is especially bad — it is impossible to discern any clear takeaway from the chart.
> > (2) Redoing these visualizations in another format such as a line or bar chart could really help readers glean the message more readily.
> > Fig. 10: the ylims on this chart make no sense. The average is reported to 3 decimals in the paper, but the chart ylims range is 400x larger. The differences between the series is almost undistinguishable from the chart. Maybe consider ylims such as (0.3, 0.35) to zoom in on the relevant portion?
>
> **A1:** We appreciate the reviewer’s detailed suggestions. We will (i) replace the radar charts in Figs. 1 and 9 with bar or line charts to improve comparison and clarity, (ii) adjust Fig. 10’s y‑axis limits and ticks to make differences discernible in the revision.
>
> > **Q2:** Minor Clarity Issues:
> > (1) Some sentences in the paper are difficult to parse and could be improved with grammar checking and rephrasing for better readability (e.g., L52-54).
> > (2) Fig. 11: what specific models generated these? Should specify in the caption?
>
> **A2:** We thank the reviewer for their constructive suggestions. To address them, we provide clarifications below and will revise the manuscript accordingly.
>
> - For the sentence in Lines 52–54, our intended meaning is as follows: In addition to learning to generate responses from images and instructions, LIT is also trained to generate instructions for given images. During instruction generation, LiT is supervised to avoid templates, special tokens, and high-frequency, low-information phrases commonly found in instructions.
> - For Fig. 11, the examples were generated by models based on LLaVA‑1.5 Vicuna‑7B.
>
> We will carefully proofread the entire paper for clarity (with particular attention to long sentences) and will include the model specification in the Fig. 11 caption.
>
>
>
>
> > **Q3:** Controlling # tokens in the loss calculation: The proposed LIT method computes the loss over more tokens (instruction + response) than the VIT baseline (response only). This raises a question about a potential confounding variable: could the observed benefits be partially attributed to simply having more tokens contributing to the gradient per sample, rather than the specific regularizing effect of predicting the instruction's content? Have you considered or can you suggest an experiment to control for this and help disentangle these two potential effects?
> > Perhaps this can be accomplished by using the same number of tokens from the system prompt, which should be uninformative.
>
> **A3:** We appreciate the reviewer’s insightful questions. Follow your valuable suggestion, we conducted a equal-token comparison that fixes the aggregate number of tokens contributing to the loss and varies only instruction informativeness. From LLaVA‑mix‑665k, we selected a 78k subset and compared model trained on: (i) an uninformative condition using fixed system and task templates $(\mathbf{X}_I^S,\mathbf{X}_I^T)$ paired with the answer $\mathbf{X}_A$; and (ii) an informative condition using the instruction with both templates removed $\mathbf{X}_I^{\backslash S,T}$, also paired with $\mathbf{X}_A$. We matched the dataset‑level instruction token counts across the two conditions, yielding comparable averages and ensuring that differences reflect instruction content rather than token count. (Response tokens $\mathbf{X}_A$ are the same across conditions, so the total token count is also matched.)
>
> | Condition (same 78k subset)     | Learning Content Used for Loss                                                                          | Avg. Tokens Contributing to Loss (Reported w/o $\mathbf{X}_A$)  |
> |-------------|--------------------------------------------------------------------------------------------|----------------------|
> | Uninformative  | system template $\mathbf{X}_I^S$, task template $\mathbf{X}_I^T$ and response $\mathbf{X}_A$       | 73.274               |
> | Informative (LiT)            | instruction excluding both the system and task templates $\mathbf{X}_I^{\backslash S,T}$ and response $\mathbf{X}_A$ | 73.273               |
>
> We then trained two model variants that differ only in the instruction‑side supervision used in the loss (informative vs. uninformative condition). The model trained on uninformative tokens failed to produce meaningful outputs in evaluation, instead replicating the template and yielding near‑zero scores on VQA, OCR, and image captioning benchmarks. These results indicate that LIT’s gains do not come from using more tokens in the loss but from informative instruction supervision, which regularize shortcut learning and places greater emphasis on the visual signal. We will incorporate the experiments into our revised manuscript.
>
> > **Q4:** Method Name: The acronym "LiT" is already widely recognized in the community for "Locked-image Tuning" (https://arxiv.org/abs/2111.07991). I am concerned that the current “LIT” name will get overshadowed by the well-established LiT method. Maybe it is worth considering an alternative name?
>
> **A4:** We appreciate the naming concern. To avoid confusion with “LiT” (Locked‑image Tuning), we will reconsider the acronym and are inclined to adopt Learning to Instruct (L2T).

---

> > ### Comment · Reviewer_S2mP · 2025-08-05
> >
> > thanks for your detailed rebuttal.
> >
> > the experiment in **A3** is great---exactly what I was looking for. and I think "L2T" is great (A4).
> >
> > all concerns addressed!

---

> > > ### Author Response · Authors · 2025-08-05
> > >
> > > Thank you for the thoughtful feedback. Your equal-token control suggestion helps us verify that the gains come from informative instruction supervision rather than token count, and your naming advice helps avoid overlap with LiT. We will incorporate these ideas and other points into the revision. Thank you again for your dedicated and valuable contribution in reviewing our paper!

---

### Official Review · Reviewer_gAGN · 2025-07-09

**Clarity:** 3
**Significance:** 2
**Originality:** 2
**Rating:** 4
**Confidence:** 4

**Summary:**

This paper primarily investigates a method to enhance the performance of multimodal large models by simultaneously computing the loss for instructions during the Supervised Fine-Tuning (SFT) phase. The authors claim that this approach acts as a form of "regularization," effectively mitigating hallucinations and improving overall performance, which is verified on many benchmarks.

**Questions:**

I kindly suggest that the authors consider adopting a broader perspective, such as examining the closed-loop interactions between SFT, rule-based RL, and RLHF, to effectively tackle challenges like hallucination and shortcut learning in VLMs. Furthermore, it would be beneficial to stay abreast of recent advancements in post-training methodologies within the language model field, with a particular emphasis on addressing key and fundamental issues.

**Ethical Concerns:**

["NO or VERY MINOR ethics concerns only"]

**Final Justification:**

Thanks for the authors' feedback. The rebuttal partly addresses my concerns, so I raise the final rating to borderline accept. The authors may consider adding the discussions in the rebuttal into the camera-ready version if the paper are accepted.

**Paper Formatting Concerns:**

No.

**Quality:**

2

**Strengths And Weaknesses:**

### Pros

The authors address an intriguing issue—how to enhance the perception capabilities of multimodal large models while reducing hallucinations. The proposed method demonstrates a notable improvement in benchmark evaluations.

### Cons
1. I find the paper's approach to be rather superficial. SFT is a highly nuanced process that demands careful attention to various aspects, including the proportion of different tasks and the quality of the response patterns (https://arxiv.org/pdf/2503.01307v1). By introducing a loss for instructions, the method effectively increases the SFT data by 50%, which is dedicated to learning how to instruct. Such a significant alteration in the SFT ratio is bound to have adverse effects, such as a potential decline in text comprehension abilities. This could be observed in evaluations like mtbench (https://arxiv.org/abs/2306.05685), wildbench (https://arxiv.org/abs/2406.04770), and fundamental text capabilities such as BBH, MMLU, and AGIEval. A decrease in language proficiency would inevitably lead to a systemic deterioration in response quality.

2. The baseline used in the article is relatively weak. I suggest testing the method on a more robust VLM baseline to determine if it still yields benefits.

---

> ### Author Rebuttal · Authors · 2025-07-31
>
> We sincerely thank the reviewer for the time and effort spent evaluating our work. We appreciate your recognition of the motivation behind our method and its effectiveness in improving perception and reducing hallucinations in multimodal large models. Below, we address your concerns in detail.
>
> > **Q1:** I find the paper's approach to be rather superficial. SFT is a highly nuanced process that demands careful attention to various aspects, including the proportion of different tasks and the quality of the response patterns (https://arxiv.org/pdf/2503.01307v1). By introducing a loss for instructions, the method effectively increases the SFT data by 50%, which is dedicated to learning how to instruct. Such a significant alteration in the SFT ratio is bound to have adverse effects, such as a potential decline in text comprehension abilities. This could be observed in evaluations like mtbench (https://arxiv.org/abs/2306.05685), wildbench (https://arxiv.org/abs/2406.04770), and fundamental text capabilities such as BBH, MMLU, and AGIEval. A decrease in language proficiency would inevitably lead to a systemic deterioration in response quality.
>
> **A1:** We thank the reviewer for this insightful comment regarding the potential impact of our method on the model's core language abilities.
>
> We respectfully argue that for an advanced Multimodal LLM, the ability to generate a relevant instruction from an image is a hallmark of deeper comprehension, not a task that should harm its language skills. We view "learning to instruct" as a synergistic form of regularization. It compels the model to more deeply process visual information to understand the context and task, thereby reducing its reliance on "shortcut learning" where it might ignore the image and guess based on language priors alone. This should lead to more robust and grounded reasoning, not weaker language abilities.
>
> To empirically address your concern, we have conducted new experiments on the text-only benchmarks you suggested. The results for the LLaVA-1.5 Vicuna-7B model are as follows:
>
>
>
> | Benchmark                 | MTbench | WildBench | MMLU  | AGIEVAL |
> |---------------------------|---------|-----------|-------|---------|
> | ViT LLaVA1.5-7B           | 5.35    | -9.40     | 49.57 | 32.62   |
> | LiT LLaVA1.5-7B           | 5.49    | -6.27     | 49.18 | 32.49   |
>
>
> As the results demonstrate, our LIT-trained model achieves comparable performance to the ViT baseline on MT-Bench, MMLU, and AGIEVAL, while showing a significant improvement on WildBench. This demonstrates that our method does not cause the hypothesized "decline in text comprehension abilities."
>
> Therefore, we believe LIT acts as an effective regularizer that enhances visual grounding and mitigates hallucination, without making a trade-off in the model's fundamental language proficiency.
>
>
>
>
> > **Q2:** The baseline used in the article is relatively weak. I suggest testing the method on a more robust VLM baseline to determine if it still yields benefits.
>
> **A2:** We thank the reviewer for their constructive suggestions. To address this, we evaluate our LiT on more robust VLM, Prism [1], which outperforms LLaVA-1.5. Prism incorporates several advanced techniques, including optimized training, image preprocessing, and fused visual backbones like SigLIP and DiNOv2, making it a more powerful baseline.
> We conducted experiments based on Prism-DINOSigLIP-Controlled-7B, applying LiT during training and comparing its performance to the standard VIT-style baseline across several benchmarks, including VQA (GQA, VizWiz, TextVQA), Localization (RefCOCO, RefCOCO+, RefCOCOg), and challenge tasks (POPE, VSR, AI2D).
>
>
> | Task         | GQA   | VizWiz | TextVQA | RefCOCO | RefCOCO+ | RefCOCOg | POPE  | VSR   | AI2D  |
> |--------------|-------|--------|---------|---------|----------|----------|-------|-------|-------|
> | ViT Prism-7B | 61.92 | 55.36  | 52.80   | 56.70   | 50.70    | 52.70    | 88.00 | 53.20 | 55.50 |
> | LiT Prism-7B | 62.62 | 57.75  | 55.60   | 66.00   | 58.90    | 62.00    | 88.50 | 61.70 | 57.10 |
>
> The results show that LiT consistently improves performance across diverse tasks. These findings highlight that LiT leads to consistent, significant improvements even with a stronger baseline like Prism. This suggests that our method has broad applicability and can enhance the performance of advanced VLMs, underscoring its effectiveness in improving visual instruction tuning. We will include the experiment and discussion in our revised manuscript.
>
>
>
> > **Q3:** I kindly suggest that the authors consider adopting a broader perspective, such as examining the closed-loop interactions between SFT, rule-based RL, and RLHF, to effectively tackle challenges like hallucination and shortcut learning in VLMs. Furthermore, it would be beneficial to stay abreast of recent advancements in post-training methodologies within the language model field, with a particular emphasis on addressing key and fundamental issues.
>
> **A3:** We sincerely thank the reviewer for their insightful and forward-looking suggestions.
>
> We agree that the interplay between SFT, RL, and RLHF is a crucial area that warrants more attention. Our work is grounded in the belief that the quality of the initial SFT stage directly influences the effectiveness of subsequent alignment [2]. Recent studies [3,4] suggest that applying RL to models already prone to hallucination or shortcut learning can reinforce these issues, resulting in confident but incorrect outputs.
>
> LIT addresses this problem at its root. By reducing shortcut learning and enhancing visual grounding during SFT, it produces a more reliable base model for RLHF. A stronger SFT foundation enables safer and more effective alignment.
>
> While recent advances have focused on RL [5] and reward modeling [6], we believe the foundational role of SFT remains important. Early-stage flaws can persist and even intensify during RL. LIT serves as a preventive measure, aiming to improve model behavior before alignment rather than correcting it afterward.
>
> We will incorporate these points into the revised manuscript to better emphasize the fundamental importance of SFT and the broader impact of our approach.
>
> [1] Prismatic VLMs: Investigating the Design Space of Visually-Conditioned Language Models. ICML 2024.
>
> [2]  Training language models to follow instructions with human feedback. NeurIPS 2022.
>
> [3] Learning Auxiliary Tasks Improves Reference-Free Hallucination Detection in Open-Domain Long-Form Generation. ACL 2025.
>
> [4] Exploring and Mitigating Shortcut Learning for Generative Large Language Models. 2024.
>
> [5] Reinforced MLLM: A Survey on RL-Based Reasoning in Multimodal Large Language Models. 2025.
>
> [6] R1-Reward: Training Multimodal Reward Model Through Stable Reinforcement Learning. 2025.

---

> > ### Author Response · Authors · 2025-08-06
> >
> > Dear Reviewer gAGN,
> >
> > Thank you again for your thoughtful feedback. As the discussion period nears its end, we’d be grateful if you could review our latest updates at your convenience.
> >
> > A brief summary for your reference:
> >
> > - Added text-only benchmarks; LiT matches baseline on MT-Bench/MMLU/AGIEval and improves on WildBench.
> >
> > - Evaluated on stronger Prism-7B baseline; consistent gains on GQA, VizWiz, TextVQA, RefCOCO/+/g, POPE, VSR, AI2D.
> >
> > - Expanded SFT–RL/RLHF discussion. LiT reduces shortcut learning and improves visual grounding during SFT, providing a stronger foundation for downstream alignment stages.
> >
> > We’d appreciate any further questions or suggestions you may have.
> >
> > Best,
> >
> > Authors of Submission7640

---

### Note · Authors · 2025-08-13

We sincerely thank all reviewers for their insightful feedback and constructive engagement throughout the discussion period. Special thanks to the Area Chair for facilitating this valuable process. We are grateful that the reviewers recognized the key strengths of our work:

- Simple, effective, and novel method to address the important problem of shortcut learning in visual instruction tuning (Reviewers S2mP, Qwwm, xAmq).

- Robust experimental validation demonstrating consistent performance gains across multiple models and benchmarks (Reviewers gAGN, S2mP).

- Clear motivation and well-explained methodology (Reviewers Qwwm, S2mP).

- Novel analysis via the "Visual Contribution" metric to quantify improvements in visual grounding (Reviewer S2mP).

During the rebuttal and discussion phases, we have addressed all major concerns through extensive experiments and clarifications, as explicitly confirmed by Reviewers S2mP, Qwwm, and xAmq. We have also thoroughly addressed all points raised by Reviewer gAGN.

In response to the feedback, we have strengthened the paper by:

- Enhancing Clarity and Scope: To better position our work, we have sharpened the distinction between our method and standard multimodal pre-training. We further elaborate on its synergy with subsequent alignment techniques like RLHF. Additionally, we have enhanced our data visualizations for improved clarity and adopted the suggested acronym L2T.
- Validating Generalizability and Robustness: We conducted new experiments to demonstrate our method's effectiveness on a stronger baseline (Prism-7B) and confirmed that it does not degrade language-only performance on benchmarks like MMLU and MT-Bench.
- Deepening the Analysis of Our Method: Through controlled ablation studies, we confirmed that performance gains originate from more informative instruction content, not merely increased token count. We also explored self-improvement cycles, showing that model-generated instructions can further boost performance, reinforcing the strength of our core contribution.

We believe these substantial demonstrations make our contribution stronger and clearer. All suggestions will be carefully incorporated to improve the quality of our work. We once again express our gratitude to all reviewers for their time and effort.

---

### Decision · Program_Chairs · 2025-09-17

**Decision:**

Accept (poster)

**Comment:**

The paper introduces Learning to Instruct (LIT), a simple yet effective modification to the visual instruction tuning objective for Multimodal Large Language Models (MLLMs). Instead of only computing the loss on the model's response, the proposed method also includes the instruction sequence in the loss calculation. This approach is designed to act as a regularizer, forcing the model to better ground its understanding in the visual input, thereby mitigating common issues like shortcut learning and hallucination. Reviewers were generally positive about the paper's clear motivation and the method's simplicity and novelty, though they initially raised significant questions regarding the method's potential side effects, the strength of the baselines, and the underlying cause of the performance gains. The authors' thorough and constructive rebuttal, which included substantial new experiments, successfully addressed nearly all of these concerns, significantly strengthening the paper and garnering strong support from the reviewers.

# Summary Of Reasons To Publish:
1) The core idea is recognized by reviewers as clever, practical, and easy to implement, addressing the important problem of shortcut learning in MLLMs.
2) The authors provided extensive new experiments during the rebuttal phase that confirmed the method's effectiveness. This includes: 2.1) Demonstrating that the method does not degrade performance on language-only benchmarks (e.g., MMLU, MT-Bench), directly addressing a key concern from Reviewer gAGN. 2.2) Validating the approach on a stronger, more recent baseline (Prism-7B), as requested by Reviewers gAGN and Qwwm, showing consistent performance improvements.
3) The author-reviewer discussion yielded significant new insights that strengthen the paper's contribution. Specifically: 3.1) A new ablation study, suggested by Reviewer S2mP, confirmed that performance gains stem from the informative content of the instruction, not merely an increased token count in the loss calculation. 3.2) An experiment inspired by Reviewer xAmq demonstrated a promising self-improvement cycle, where model-generated instructions could be used to further boost performance.

# Summary Of Suggested Revisions:
To ensure the final version reflects the significant improvements made during the review process, the authors are expected to incorporate the following points discussed and agreed upon in their rebuttal:
1) The final paper must include the comprehensive results from the new experiments conducted for the rebuttal, including the language-only benchmark evaluations, the stronger Prism-7B baseline results, the equal-token control experiment, and the self-improvement cycle experiment.
2) As agreed with Reviewer S2mP, the problematic radar charts should be replaced with clearer formats like bar or line charts to improve interpretability.
3) To avoid confusion with a prior method, the acronym should be changed from "LIT" to "L2T" (Learning to Instruct), as suggested by Reviewer S2mP and accepted by the authors.
4) The manuscript should be updated to include the clarifications provided to Reviewer Qwwm, particularly the detailed distinction between this fine-tuning technique and standard multimodal pre-training.
5) The discussion should incorporate the acknowledged limitations regarding the ad-hoc nature of template removal, as raised by Reviewer xAmq, and frame it as a direction for future work.